# The Impact of Coreset Selection on Spurious Correlations and Group Robustness

**Amaya Dharmasiri**[1]     **William Yang**[1]     **Polina Kirichenko**[2]

**Lydia T. Liu**[1]     **Olga Russakovsky**[1]

[1]Princeton University     [2]FAIR at Meta

{amayadharmasiri, williamyang, ltliu, olgarus}@princeton.edu,
polkirichenko@meta.com

## Abstract

Coreset selection methods have shown promise in reducing the training data size while maintaining model performance for data-efficient machine learning. However, as many datasets suffer from biases that cause models to learn spurious correlations instead of causal features, it is important to understand whether and how dataset reduction methods may perpetuate, amplify, or mitigate these biases. In this work, we conduct the first comprehensive analysis of the implications of data selection on the spurious bias levels of the selected coresets and the robustness of downstream models trained on them. We use an extensive experimental setting spanning ten different spurious correlations benchmarks, five score metrics to characterize sample importance/ difficulty, and five data selection policies across a broad range of coreset sizes. Thereby, we unravel a series of nontrivial nuances in interactions between sample difficulty and bias alignment, as well as dataset bias and resultant model robustness. For example, we find that selecting coresets using embedding-based sample characterization scores runs a comparatively lower risk of inadvertently exacerbating bias than selecting using characterizations based on learning dynamics. Most importantly, our analysis reveals that although some coreset selection methods could achieve lower bias levels by prioritizing difficult samples, they do not reliably guarantee downstream robustness.[1]

## 1   Introduction

The recent success of over-parameterized models is largely driven by the vast scale of available data [1, 2]. However, as datasets grow to an unprecedented scale, reaching billions of images [3, 4], training deep models on them demands enormous computational resources. Moreover, web-scale datasets tend to be noisy, with many samples of low informativeness and quality [5, 6]. These concerns have driven interest in selecting high-quality, informative subsets of data, also known as *coresets*. Additionally, there is a growing interest in understanding the role of data in the model's generalization, characterizing sample-level learning dynamics, and identifying examples with the highest influence on the model's predictions [7–9]. Building on this broader goal of data-centric machine learning, coreset selection [10–16] aims to identify a small subset of training data that preserves the original model performance while enhancing training efficiency.

---

[1]See https://github.com/princetonvisualai/Robustness-impacts-of-coreset-selection for results and code.

However, real-world datasets often contain biases that cause trained models to rely on **spurious features** instead of causal features [17–22]. This reliance can undermine model generalization, especially when deploying models in environments where these spurious correlations do not hold [23, 24]. This issue could be further exacerbated if models are trained on subsets of such data, as the subset selection process can amplify the impact of spurious features, leading to decreased robustness and fairness in model performance. While there has been significant progress in coreset selection methods, they are predominantly evaluated on datasets like CIFAR [25] or ImageNet [26] using average accuracy as the metric [10, 12, 11, 13, 16, 27, 15, 14]; the downstream impacts of models trained on coresets of data with complicated spurious features remain unexplored.

We begin with a quick preview of some of our results to demonstrate these potential concerns. Here we consider the Waterbirds [20] dataset, where labels (waterbirds vs landbirds) co-occur with the backgrounds (land or water), forming four distinct *subgroups*. In the training set, 95% of the waterbirds appear on water backgrounds, and 95% of landbirds appear on land backgrounds, making the backgrounds a spurious feature. This association can lead the model to predict based on the background rather than on the bird, incurring disproportionately high errors for waterbirds on land, and landbirds on water [17, 28].

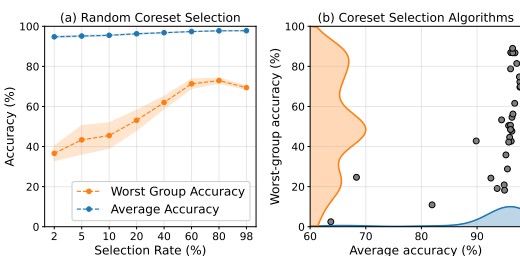

Figure 1: *Left:* Performance of randomly selected coresets from Waterbirds [20] reveals an increasing discrepancy between worst-group accuracy and average accuracy as coreset size decreases. *Right:* Distribution of average and worst-group accuracy for current coreset selection algorithms shows large variation in worst-group accuracies even for similar average accuracies.

In Figure 1 *Left*, we demonstrate this disproportionate effect as the size of the coreset changes. We compare the **worst-group accuracy** (the lowest accuracy among the four subgroups [20]) and the average test accuracy of a model trained on coresets of different selection ratios. The coresets are selected using uniform random sampling (or random coreset selection), which is considered a strong baseline [11, 10]. Although classifiers continue to achieve high average accuracy with very small coresets, their robustness as measured by worst-group accuracy declines substantially. Hence, we highlight the importance of understanding how different coreset sizes impact the group robustness of resulting models.

In Figure 1 *Right* we conside more complex coreset selection methods and plot the distribution of average and worst-group accuracies for models trained on 10% coresets. Each point corresponds to the combination of one sample characterization score [27, 15, 14, 13, 12] and a selection policy [11–13]. We show that different coresets with comparable downstream average accuracies can result in drastically different levels of group-robustness. Hence, we emphasize the crucial need for a deeper evaluation to understand how different strategies for coreset selection impact model robustness and how to select coresets in the presence of spurious correlations.

We conduct the first systematic, large-scale empirical study of how coreset selection impacts dataset bias and downstream group-robustness. In our analysis, we evaluate five different sample characterization scores spanning two families (learning-based and embedding-based methods) [27, 15, 14, 13, 12] and five different selection policies from the coreset selection literature [13, 11, 12], across ten diverse datasets with spurious correlations spanning both visual and natural language classification tasks [29, 20, 30–35]. We show that coreset selection on datasets with spurious correlations is highly nuanced, where the induced bias of the coreset and downstream group-robustness are strongly influenced by the scoring method, selection policy, and coreset size. For example, we show that **sample characterizations scores based on feature embeddings [13, 12] run a lower risk of inadvertently exacerbating bias when used for selecting coresets compared to scores based on learning dynamics [27, 15, 14]**, and that **lower bias levels of coresets does not reliably guarantee downstream robustness**. Most importantly, we show that **special considerations need to be made when the coreset size is very small**, since there is a unique risk of highly prototypical coresets reaching high average performance while obscuring their low group-robustness.

# 2   Related Work

**Data selection.** Within the broader goal of data-efficient learning, multiple research threads diverge in nuanced ways. A line of research synonymously referred to as data pruning seeks to remove uninformative samples from a larger dataset [13, 11, 16]), whereas coreset selection methods try to find a small subset of data that is representative of the whole dataset [10]. Data selection methods stemming from data attribution searches for the most important samples using influence functions [36, 37], shapley contribution of each sample to the model performance [38], or the ability of the training data to predict the model behavior [39]. In the coreset selection literature, less difficult (easy) samples are often synonymous with redundant and less informative, and the prescribed selection policy is to retain the hardest samples.

Recent work uncovered that the optimal policy for selecting coresets depends on the dataset size and the coreset size: selecting the hardest samples can be suboptimal for learning an accurate classifier, especially when the coreset size is very small [13]. Furthermore, there seem to be many sources of hardness/difficulty for samples [40], and not all of them convey important information for training generalizable classifiers. Given that many sample characterization scores are strongly correlated [41], we investigate whether the concept of difficulty used in data selection is tied to spurious correlations.

Some methods, in contrast, avoid explicit importance scores and instead leverage distributional properties of the dataset [42, 43]. Active learning and dynamic pruning methods share similar flavors to data selection but are out of the scope of our consideration.

**Spurious correlations and group robustness.** Mitigating spurious correlations is approached broadly in two ways: model interventions [20, 44] and data interventions [45–47]. When group labels are available, simple data balancing has been shown to achieve competitive group-robustness [45]. A notable limitation of these methods is the frequent unavailability, ambiguity, or high cost of obtaining sample-level group labels in real-world settings. Therefore, the bulk of our analysis focuses on addressing spurious correlation in the absence of group labels.

Concurrently, there is research focused on identifying minority subpopulations within a dataset by quantifying their influence on the training process and reweighting them to mitigate downstream bias [48, 7, 49, 50]. These methods use loss and error signals, much like certain coreset selection strategies, to identify minority groups. Bias discovery aims to discover consistent patterns in the data to expose previously unknown biases and corresponding error patterns [51, 32, 52]. However, data interventions geared toward group robustness also run the risk of compromising the model's natural accuracy on skewed datasets [53].

**Data selection meets group-robustness.** Pruning out instances of certain groups or classes when datasets are imbalanced, also coined "subsampling," aims to reduce bias and improve robustness of the downstream classifiers [54]. This is in line with previous work on data interventions for spurious correlation mitigation. A recent study [55] hints that data pruning may mitigate distributional bias in trained models- however it was limited to one data selection method: EL2N [27]. A concurrent work [56] has explored how to prune out certain training samples to mitigate spurious correlations, in a setting where the spurious signal is relatively weaker. Distributionally robust data pruning (DRoP) [57] emulates this behavior at the class-level, retaining more samples from the more difficult classes. However, we reiterate the difficulty of our setting compared to this, since group labels are often unavailable, and there could be unknown spurious correlations.

In this work, we seek to bridge this knowledge gap by conducting an extensive study of how coreset selection methods can impact spurious biases of datasets and classifiers. We place this work within a broader effort to systematically analyze existing techniques and datasets, aiming to extract key insights that advance understanding and inform future research [58–60].

# 3   Experimental setup

In this section, we formally define spurious correlations, formulate the coreset selection problem (including the sample scoring methods and selection policies we consider in our analysis), and elaborate on our experimental setup.

**Quantifying the strength of spurious correlations.** Spurious correlation is a type of *bias* that occurs when the dataset contains a spurious feature that is predictive of the target label on the training set, but not necessarily so on the test set. We assume that each sample $x_i \in \mathcal{X}$ is associated with a spurious feature $a_i \in \mathcal{A}$. In our analysis, we quantify *bias*, $B$ as a statistical relationship between the target label (class) $y$ and a particular spurious feature $a$. Bias in a dataset is considered present between the class $y$ and the spurious feature $a$ if $B(y, a) > 1$. To characterize the overall bias in the dataset, we define the ***bias level*** as follows:

$$\text{Bias level} = \max_{y,a} B(y, a) \quad \text{where} \quad B(y, a) = \frac{P(a|y)}{P(a)} \tag{1}$$

This represents the highest degree of dependency between any class $y$ and spurious feature $a$ within the dataset. For example, in the Waterbirds dataset [20], there are two classes: landbirds and waterbirds, as well as two spurious features: land-backgrounds and water-backgrounds. Using the dependency measure that we defined above, we can categorize samples into two distinct categories:

- **Bias-aligning:** Pairs $(y, a)$ for which $B(y, a) > 1$, i.e., the spurious feature $a$ is positively associated with the class $y$ are Bias-aligning groups. The Waterbirds dataset [20] has two bias-aligning groups: waterbirds on water backgrounds and landbirds on land backgrounds. Corresponding individual samples from either of these two groups are *bias-aligning samples*.

- **Bias-conflicting:** Pairs $(y, a)$ for which $B(y, a) < 1$ are Bias-conflicting groups. The spurious feature $a$ appears less frequently in class $y$, suggesting a negative association and potentially a harder recognition challenge. The Waterbirds dataset [20] has two bias-conflicting groups: waterbirds on land backgrounds and landbirds on water backgrounds. Corresponding individual samples from either of these two groups are *bias-conflicting samples*.

**Coreset selection.** Given a large training set $\mathcal{T} = \{(x_i, y_i)\}_{i=1}^{|\mathcal{T}|}$, coreset selection aims to find a subset $S \subset \mathcal{T}$ with a desired $|S|$, so that the model $\theta^S$ trained on $S$ has close generalization performance to the model $\theta^{\mathcal{T}}$ trained on the whole training set $\mathcal{T}$. We formalize this process into two stages: *sample characterization*, in which individual scores are assigned to each sample to quantify its importance or difficulty , and *sample selection*, where a particular policy defines how these scores guide the construction of the final coreset. The scoring function $f(\phi, x_i)$ that assigns sample-level scores is parameterized by $\phi$, and can be characterized into two groups:

- **Learning-based** scores train a surrogate model $\phi^{\mathcal{T}}$ on the entire train set $\mathcal{T}$ and use it to parameterize the score assignment [61]. We investigate three such scores in our analysis: **EL2N** score [27] is calculated as $A(\phi^{\mathcal{T}}, x_i) = \|\sigma[\phi^{\mathcal{T}}(x_i)] - \hat{y}_i\|_2$ where $\sigma$ denotes the softmax function on output logits and $\hat{y}$ is the groundtruth label in one-hot. **Uncertainty** score [14] is the entropy of class prediction probabilities $\phi^{\mathcal{T}}(x_i)$. **Forgetting** score [15] is defined as the number of times $(x_i, y_i)$ is correctly learned and subsequently forgotten during the training of the surrogate model $\phi^{\mathcal{T}}$.

- **Embedding-based scores** use a pretrained feature extractor $\phi^*$ to derive feature embeddings $\mathcal{Z}$ for $\mathcal{T}$, and a distance function is used to estimate the informativeness/uniqueness of a sample in the embedding space. We use the following embedding-based scores in our analysis: Self-supervised (**SelfSup**) score [13] runs k-means clustering on the latent space and scores each sample with the distance to its nearest centroid. Supervised prototypes (**SupProto**) score [12] calculates the center of each class in the latent space and uses the distance of each sample to its corresponding class center as its score.

We consider several **sample selection policies** including: the **Difficult** policy, which selects the highest scoring samples for the coreset, the **Easy** policy, which selects the lowest scoring samples [13], the **Median** policy, which selects samples closest to the median of the score distribution [12], and the **Stratified** policy [11], which construct 50 bins ranking from lowest to highest scores and sample randomly from each bin based on the data budget allocated for each bin. We implement the **Random** selection policy as a baseline. Additionally, we implement an oracle selection policy **Random-Groupbalanced** which picks an equal number of random samples from each group to create a group-balanced coreset. (The extent of group-balancing is limited by the number of total samples present from each group, since we do not oversample.) **R-Gbal** acts as an upper bound since

it utilizes both class labels and spurious feature labels to identify the groups, whereas we assume that only class labels are available in our setting.

Existing data selection methods can introduce class imbalance [57, 13], leading to challenges with model training, especially for small dataset sizes (e.g., some of the classes could be completely excluded). To disentangle potential class imbalances from the effects of spurious correlations, we perform **class balancing** by selecting equal proportions of samples from each class when constructing the coreset (up to the limits posed by the original dataset; see Appendix for more details).

**Datasets.** We run our analysis on 10 different datasets containing known spurious correlations. All datasets are designed for classification tasks, and contain different types of spurious features: *backgrounds* with Waterbirds [20], Metashift [31], Nico-spurious [32, 62], Urbancars-B [30], *co-occuring objects* with Urbancars-C [30], Civilcomments [34], MultiNLI [20, 35], *visual features* with cMNIST [29], Celeb-A hair [33], and *a mixture of multiple spurious features* with Urbancars-A [30]. (See Appendix A for full dataset descriptions and their Bias-levels.)

**Analysis pipeline.** The pipeline of our analysis for a dataset with known spurious correlations is as follows. First, we assign **sample-level scores** to the dataset using a sample characterization score. Next, we select coresets of a desired size by applying a **sample selection policy** on the samples as characterized by the scores calculated in the first step. We use the *bias level* metric as introduced in Equation 1 to quantify the strength of the spurious correlations of the resulting coreset. Finally, we train a **downstream classifier** on the selected coresets.

We keep the number of training iterations fixed across our experiments by setting the number of training epochs to $N/s$ where $N$ is the number of training epochs on the full dataset, and $s$ is the coreset size as a fraction of the full dataset. We use models initialized with Imagenet-1K [26] pretrained weights for the visual datasets: ResNet18 [63] for cMNIST, and ResNet50 for others. NLP datasets were classified using BERT [64] model pretrained on Book Corpus and English Wikipedia data. The same models and initializations were used as surrogate models to calculate characterization scores. (Please refer to Appendix B for complete training details.)

We evaluate the average performance of the downstream classifier using **Average accuracy**, which assumes the test set has a similar group distribution as the original training set. This is done by computing group accuracies and re-weighting them according to their proportions in the original train dataset [20, 46]. Lastly and most importantly, we measure the group-robustness of the downstream classifier using **worst-group accuracy**, which is the minimum of the individual group accuracies. A high **worst-group accuracy** reflects that the model has learnt a solution that is robust to the known spurious correlation of the dataset.

# 4 Empirical analysis

## 4.1 Embedding-based characterization scores run a lower risk of inadvertently exacerbating bias compared to learning-based scores

We begin by investigating whether sample characterization scores used by coreset selection algorithms are predictive of bias-conflicting samples since coreset selection methods may inadvertently amplify bias in the subsampled dataset. We do this by (1) determining whether sample characterization *scores* can be used directly to classify a dataset into bias-aligning versus bias-conflicting samples and (2) examining the level of bias in coresets that are selected under different sample selection *policies*. Later in section 4.2, we will examine the effect that this selected data has on the downstream group-robustness of the resulting classifier.

**Identifying bias-conflicting samples using characterization scores.** We utilize coreset selection scores as predictive signals for bias-conflicting samples using average precision (AP). Figure 2 reveals that while most learning-based scores are informative for detecting bias-conflicting samples, embedding-based scores do not appear to be.

In more detail, two learning-based scoring methods, EL2N [27] and Uncertainty [14] are substantially better than the random baseline at classifying bias-conflicting vs bias-aligning samples across all 5 datasets, suggesting that coresets selected using these scores run the risk of systematically excluding

certain groups, inadvertently exacerbating bias levels. The third learning-based scoring method, Forgetting [15], is more nuanced, appearing to contain bias information on Waterbirds (23.9% AP vs 4.7% AP for the random baseline) and Urbancars-C (13.8% vs 5.0%) but comparatively little on the other datasets (e.g., on MetaShift 35.9% AP vs 30.4% AP for the random baseline).

Most striking though, ***embedding-based scoring methods provide very little information towards identifying bias-conflicting samples on real-world datasets***. Concretely, both SelfSup [13] and SupProto [12] achieve near-random AP on Urbancars-C, Metashift, and Civlcomments datasets. We further finetune these embeddings on each of these datasets for 20 epochs and the pattern persists: SupProto achieves on 5.2% AP (6.8% with finetuning) vs 5.0% AP for the random baseline on Urbancars, 35.0% AP (34.5% with finetuning) vs 30.4% on Metashift and 37.3% AP (45.8%) vs 38.2% on Civlcomments. Only on the very simple cMNIST and to a lesser extent on Waterbirds, where the discriminative features are simple, do these methods provide a substantial signal. As we will further see below, this suggests that ***embedding-based coreset selection methods (even after finetuning) pose lower risk of bias exacerbation in real-world datasets, compared to learning-based methods.***

**Coreset bias levels under different selection policies.** We directly evaluate the bias levels using Equation 1 of the coresets selected using different sample characterization scores and policies. As expected, embedding-based scoring methods produce coresets with similar bias levels regardless of the selection policy: for example, selecting 10% of the data using the Difficult, Easy, Median and Stratified policies on SelfSup scores results in bias levels between 1.89-2.00 on the Waterbirds dataset (1.87 is the bias on a random coreset of similar size), and 1.92-1.93 on Urbancars-c (1.92 is the bias on a random coreset of similar size).

Learning-based scores, on the other hand, unsurprisingly provide stronger indicators for bias, and in return, the bias levels of the resultant coresets vary largely with the executed selection policy. This can be seen in the left column of Figure 3 (which we will discuss in more detail in the next section); for example, selecting 10% of the Urbancars-C data with learning-based EL2N scores varies between bias level of 1.5 using Difficult, and 2.0 using Easy compared to the bias level of 1.9 on the random coreset. Therefore, care needs to be given when using learning-based coreset selection scores, especially in the presence of unknown spurious biases in the data. Hence, we focus more on the inadvertent risks of learning based selections in the subsequent sections.

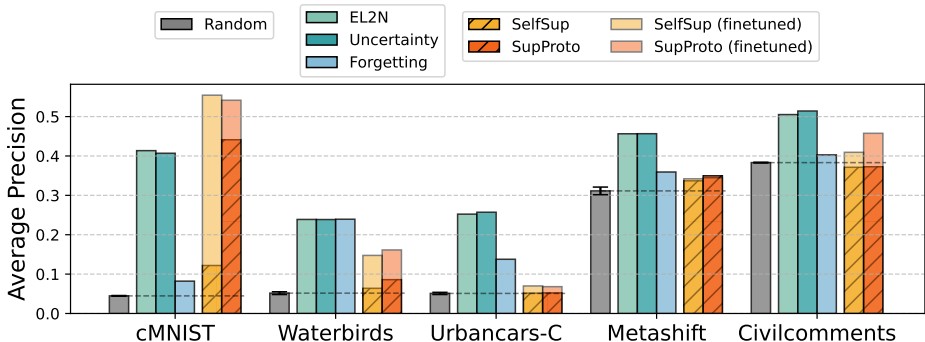

Figure 2: **Classifying bias-conflicting samples using characterization scores.** We measure the Average Precision of three learning-based methods (EL2N [27], Uncertainty [14] and Forgetting [15]) and two embedding-based methods (SelfSup [13] and SupProto [12]) at classifying bias-conflicting vs bias-aligning samples across 5 datasets. On the more challenging real-world datasets (Urbancars-C [30], Metashift [31], and Civlcomments [34]), embedding-based methods do not appear to order the samples according to their bias levels (i.e., have near-random AP); even finetuning these embeddings (depicted by the shaded bars) does not change these findings. (Please refer Appendix C.1 for more results on other datasets)

## 4.2 Coreset bias level is not a consistent indicator of downstream robustness

Our findings in Section 4.1 demonstrate that different coreset selection methods can lead to coresets with different bias levels compared to the corresponding full datasets. Inspired by the extensive literature that suggests that constructing group-balanced training sets can improve group-robustness [45–

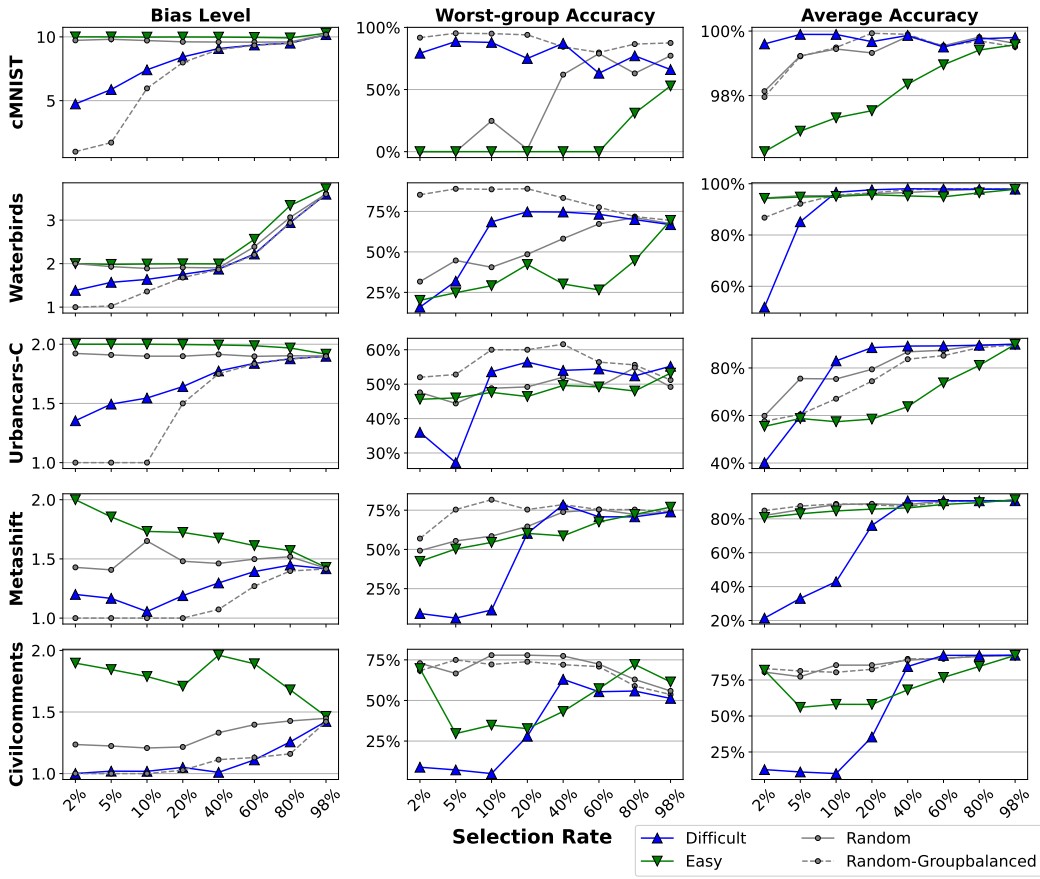

Figure 3: **Data bias and classifier accuracies for Difficult (highest-scoring) and Easy (lowest-scoring) scoring samples using EL2N scores.** Selecting the Difficult samples typically results in less biased coresets and corresponding more robust (highest worst-group accuracy) classifiers than Easy samples. The Difficult samples also tend to be more robust than those with Random selection. However, for small coreset sizes, we see a drop in average and worst-group accuracies for Difficult samples, which we examine further in Section 4.3. (Please refer Appendix C.2 for more results on other datasets.)

47, 65] of models, we examine whether coreset selection methods can correspondingly help (or hurt!) classifier group-robustness.

In Figure 3, we show the coreset bias-level (left column) and worst-group and average accuracy of a classifier trained on those coresets (middle and right columns) across five datasets (rows) with different selection policies (colors) using the EL2N [55] scores. Selection rate (x-axes) indicates the size of the coreset as a percentage of the full dataset. In the leftmost column, we observe that coresets selected using the Difficult policies consistently have a lower bias level compared to those selected using the Easy or Random policies. Based on many prior works [45–47, 65], one would expect Difficult coresets to result in classifiers with higher group-robustness (lower worst-group accuracy). However, this is only partially true. In the middle column of Figure 3, we see that **the Difficult coresets lead to more robust classifiers (classifiers with higher worst-group accuracy) but only when the coreset size is "sufficiently large."** (What constitutes "sufficiently large" appears to vary empirically between datasets: e.g., greater than about 10% of the data size for Waterbirds and Urbancars, and greater than 40% for Metashift and Civilcomments.)

In Table 1, we report the worst-group accuracy for 40% coresets selected using EL2N [27] as well as SelfSup [13] across all 10 datasets. The best-performing configurations (bolded values) in terms of worst-group accuracy predominantly correspond to Difficult selections using EL2N, implying that at 40% selection rate, lower bias levels can help models achieve higher-group robustness.

| Dataset | Baselines | | EL2N [55] scores | | | | SelfSup [13] scores | | | |
|---|---|---|---|---|---|---|---|---|---|---|
| | R | R-Gbal | Diff | Strat | Med | Eas | Diff | Strat | Med | Eas |
| cMNIST [29] | 62.0 | 84.4 | **87.2** | 74.0 | 0.0 | 0.0 | 83.0 | 83.5 | 44.0 | 0.0 |
| Waterbirds [20] | 58.2 | 83.3 | **74.6** | 73.1 | 50.5 | 30.2 | 50.9 | 51.0 | 69.2 | 37.1 |
| Urbancars-C [30] | 52.0 | 61.6 | 54.0 | 51.2 | _46.4_ | _49.6_ | 42.0 | 48.4 | **54.4** | 44.0 |
| Metashift [31] | 73.8 | 78.5 | **78.5** | 63.1 | 66.5 | 58.6 | 73.3 | 69.2 | 74.3 | 56.9 |
| Civilcomments [34] | 77.4 | 72.0 | 63.0 | 51.7 | 63.9 | _43.2_ | 77.8 | 78.4 | 70.3 | **79.3** |
| Nico-spurious [32, 62] | 44.0 | 40.0 | **44.0** | 44.0 | 34.0 | 32.0 | 44.0 | 16.0 | 34.0 | 36.0 |
| Urbancars-B [30] | 44.8 | 64.0 | **52.0** | 44.8 | 26.4 | 15.6 | 43.2 | 48.0 | 41.6 | 26.4 |
| Urbancars-A [30] | 16.8 | 50.4 | **23.2** | 19.2 | 6.4 | 7.2 | 11.2 | 17.6 | 20.8 | 9.6 |
| MultiNLI [20, 35] | 60.5 | 57.6 | _46.0_ | **65.6** | 61.8 | 46.0 | 55.2 | 60.8 | 55.7 | 58.3 |
| Celeb-A hair [33] | 58.9 | 71.1 | 38.3 | 47.8 | 68.9 | **79.0** | 41.7 | 66.1 | 63.3 | 78.9 |

Table 1: **Worst-group accuracies for different selection policies at 40% selection rate.** The _small_ values correspond to models with a catastrophic drop in average accuracy (10.0 worse than the corresponding random baseline). In general, Difficult selection policies with EL2N scores yield robust classifiers. Across the results for SelfSup scores, no one policy stands out consistently. (Please refer Appendix C.2 for more results)

On the flip side, as shown in the middle columns of Figure 3, when the coreset sizes get smaller, the robustness of models for Difficult coresets becomes unintuitively low, despite the bias levels being the lowest out of all policies. This behaviour could be partially explained by the corresponding drop in average accuracy (third column of Figure 1). Many state-of-the-art coreset selection methods are known to result in a catastrophic drop in average accuracy for small coreset sizes [11, 66]. Recent works have attributed this phenomenon to low data coverage due to selecting only the most difficult samples [11], or the models overfitting on very specific hard examples [13], leading to poor generalization. However, the reality is more nuanced, and we examine this further in the following section.

## 4.3 Very small coresets require special considerations

We established that Difficult selection with learning based scores yields the least biased coresets. However, it does not translate to improved group robustness for small coreset sizes. In this section, we perform a deeper analysis of this discrepancy to understand the nuances of the small data regime.

**Coresets of Easy samples may yield better average performance but offer no robustness benefits.** One proposed solution for the catastrophic drop in average accuracy for very small coresets is selecting the easiest samples for the coreset [13]. As shown in the rightmost column of Figure 3, for all datasets except cMNIST, Easy selection surpasses Difficult in average accuracy for very small selection rates. However, the corresponding worst-group accuracy for Easy policy, although higher than that of Difficult policy, remains well below simple Random selection. This reinforces our prior finding that easy samples are predominantly bias-aligned, resulting in a highly biased coreset, eventually degrading group-robustness. At very small coreset sizes, Easy selection policy often yields high average accuracy, masking the fact that it systematically excludes bias-conflicting samples and still leads to worse group-robustness.

**Group-balancing alone does not guarantee improved robustness; difficulty of the samples play an important role.** In the regime of very small coreset sizes, we observe that Difficult selection, although yeilds the lowest bias-levels, does not guarentee group-robustness. This observation is in direct contradiction with many prior works that established the benefits of group-balanced training sets (equal number of samples from all individual bias-aligning and bias-conflicting groups) on group-robustness in the presence of spurious correlations. To explain this apparent contradiction, we hypothesize that not all group-balanced coresets have similar robustness benefits, but another factor: _the sample difficulty within each group_ could have a conflating impact.

To test this hypothesis, we implement a few selection policies by incorporating EL2N difficulty scores with oracle information about which group each sample belongs to. R-Gbal baseline selects a group-balanced coreset in which the target number of samples from within each group is selected randomly. Additionally, we implement Diff-Gbal and Eas-Gbal selection policies, which select the same _number of samples from each group_ as R-Gbal selection, but apply the corresponding policies

Difficult and Easy within the samples of each group. (The level of group-balancing is limited by the number of samples from each group in the overall dataset and the coreset size). The resultant worst-group accuracies of the classifiers trained on these selections are shown in Figure 4. We observe a clear difference between R-Gbal and Diff-Gbal. Even though all policies have equal bias levels at a given selection rate, prioritizing the most difficult samples from each group (Diff-Gbal) leads to worse robustness compared to uniformly random (R-Gbal) or easiest (Eas-Gbal) samples.

An array of scoring methods that approximate sample difficulty have been repurposed to identify bias-conflicting examples, in settings where annotations of spurious features are absent [48, 67, 68, 44, 69, 70]. Bias-conflicting samples thus identified are in turn used to construct more group-balanced training sets, or incorporated into model interventions such as weighted optimization [48] to combat the models from learning spurious correlations. Our observation that **"*group balancing alone doesn't guarantee a model's group-robustness; the difficulty of samples within each group play an important role*"** raises an interesting caveat on this line of methods for discovering bias-conflicting samples for group-robustness.

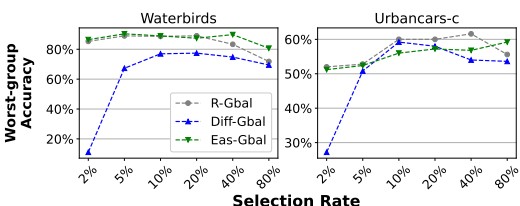

Figure 4: **Worst-group accuracies for Waterbirds [20] and Urbancars-C [30] using group-balanced coresets.** Although all methods have equal bias-levels, selecting the most difficult samples from each group results in comparable or worse group-robustness.

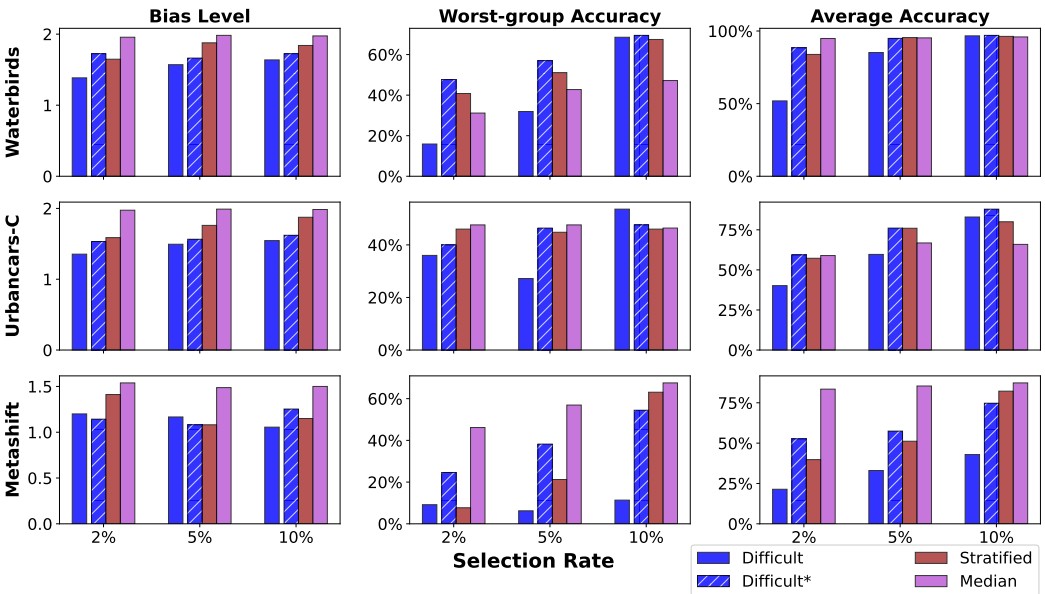

Figure 5: **Effect of excluding most difficult bias-conflicting samples in the small data regime.** Median and Stratified selection policies perform better than Difficult selection on worst-group accuracy for small coreset sizes, although the slightly modified Difficult* selection strategy discussed in Section 4.3 mitigates the difference. (Please refer Appendix C.3 for more results on other datasets)

**Trading off most difficult bias-conflicting samples to improve robustness.** Learning based scores that characterize the difficulty of samples assign high values to rare and unique instances that carry fine-grained information about the particular class. However, they are also known to assign high scores to mislabeled samples [27, 71, 72] or samples that are too far out-of-distribution (OOD) to be useful for learning the target function [7, 73, 40]. To avoid the negative impact of such noisy samples in very small coresets, excluding a small number of highest scoring (most difficult) samples [27], selecting samples from different levels of difficulty [11], or simply selecting samples whose difficulty is close to the dataset median [12] has been proposed. Here we investigate how such selection strategies impact the downstream group-robustness.

We implement Difficult* selection policy by excluding a small percentage (3%) of the highest scoring samples as a simple heuristic. Difficult*, Stratified, and Median selection policies therefore respectively correspond to the three aforementioned mitigation methods [27, 11, 12]. Bias-levels, worst-group, and average accuracies for these selection policies in the small data regime are shown in Figure 5. Although Difficult*, Stratified, and Median progressively increase the bias-level of the coresets compared to Difficult selection, they result in improved robustness (worst-group accuracy) in cases where Difficult policy has catastrophically low robustness. (e.g: at 2% and 5% on Waterbirds and Urbancars-C, and 2%, 5%, 10% on Metashift) However, we argue that these strategies are still far from being one-size-fits-all solutions; (e.g: at 10% on Waterbirds and Urbancars-C, Median policy resulted in exacerbated bias-levels, harming the robustness compared to Difficult)

# 5   Conclusions

We conducted the first systematic and multidimensional empirical study of how coreset selection impacts dataset bias and downstream group robustness. Through consistent patterns across our extensive experimental setup, we expose interesting nuances to the common knowledge that "bias-conflicting samples are difficult to learn" and "lower bias in datasets leads to higher robustness", in the context of selecting coresets and training models on coresets.

Our analysis points to a heuristic in how we can do coreset selection when access to detailed annotations of spurious features is unavailable: select coresets prioritizing the most difficult/rare/non-prototypical samples, and ensure that the coreset is sufficiently large to reach comparable or higher performance than a uniform random coreset of similar size. Thereby, alleviate the risk of systematically excluding minority samples and avoid the pitfalls of the small-data regime.

---

**Algorithm 1** Coreset Selection ensuring Group-Robustness (Group information is unavailable)

---

**Given:** A training + validation set with class labels but no spurious annotations, and a desired coreset size range $[N_{\min}, N_{\max}]$ (dictated by application constraints).
**Goal:** Find the smallest coreset likely to yield high accuracy and group robustness on a test set with the same target classes and unknown group labels.
**Method:**
    **Initialize:** Set coreset size $N = N_{\min}$.
    **while** $N \leq N_{\max}$ **do**
        Select two coresets: $S_{N,\text{difficult}}$ using the **difficult** policy, $S_{N,\text{random}}$ using the **random** policy
        Train models: $M_{N,\text{difficult}}$ on $S_{N,\text{difficult}}$, $M_{N,\text{random}}$ on $S_{N,\text{random}}$
        **if** avg accuracy($M_{N,\text{difficult}}$) $\approx$ avg accuracy($M_{N,\text{random}}$) **then**
            **return** $S_{N,\text{difficult}}$ $\triangleright$ $N$ is "sufficiently large," difficult coreset likely yeilds high robustness.
        **else**
            Increase $N$       $\triangleright$ $N$ is in the "small-data regime." $S_{N,\text{difficult}}$ may not ensure robustness.
        **end if**
    **end while**
    **Fallback:** If $N_{\max}$ is reached, use Stratified or Median policies.     $\triangleright$ These strategies, while not one-size-fits-all, are more reliable in the small-data regime.

---

**Broader impacts and limitations.** Ensuring high model performance across different groups of the test distribution is motivated by notions of group fairness, which is a crucial topic in the discourse of societal implications of machine learning. We note that group robustness, as formalized in this work, is a meaningful yet coarse and limited notion of fairness. We conducted our analysis on a set of extensive yet finite set of datasets and coreset selection methods. We acknowledge that they do not fully represent all types of spurious correlations present in the real world. Nevertheless, we hope that our findings will aid future research in building more rigorous and fair data selection methods.

# Acknowledgments

This material is based upon work supported by the National Science Foundation under Grants No 2112562 and 2145198. Any opinions, findings, and conclusions or recommendations expressed in this material are those of the author(s) and do not necessarily reflect the views of the National Science Foundation. All experiments, data collection, and processing activities were conducted at Princeton University. Meta was involved solely in an advisory role and no experiments, data collection or processing activities were conducted on Meta infrastructure. We thank Angelina Wang for insightful discussions during the early stages of the project.

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

# Appendix

This appendix includes further details on the experiment setup and analysis of the paper "The Impact of Coreset Selection on Spurious Correlations and Group Robustness".

Our results and code are publicly available here

# A   Datasets, Characterization scores, and Policies

**cMNIST (c-Mn)** [29]- A simple synthetic version of the MNIST [25] dataset where colors have been added to the images of the numbers. Each digit is spuriously correlated with a specific color.

**Waterbirds (WB)** [20]- constructed by placing images from the Caltech-UCSD Birds-200-2011 [74] dataset over backgrounds from the Places [75] dataset. The task is to classify whether a bird is a landbird or a waterbird, where the spurious attribute is the background (water or land).

**Urbancars** [30]- The task is the classification of car images into urban cars and country cars. There are 2 different spurious attributes that result in three different sub-datasets. **Urbancars-C (UC-C)** has a co-occurring object from urban and country contexts as the spurious feature, whereas **Urbancars-B (UC-B)** has backgrounds as the spurious feature. **Urbancars-A (UC-A)** has both spurious features resulting in 8 different subgroups.

**Metashift (MSh)** [31] MetaShift is a general method of creating image datasets from the Visual Genome project [76]. We use the Cat vs. Dog dataset, where the spurious attribute is the image background. Cats and more likely to be indoors, and dogs are more likely to be outdoors. We use the "unmixed" version according to the original implementation.

**Nicospurious (Nic-S)** [62, 32] NICO++ is a large-scale benchmark for domain generalization. We only use their training dataset, which consists of 60 classes and 6 common attributes (autumn, dim, grass, outdoor, rock, water). To transform this dataset into the spurious correlation setting, we use the method followed by [32]

**Civilcomments (CC)** [34]- a text classification task where the goal is to classify a given comment as "toxic" or "neutral". Following prior works [45] we use the coarse version of the dataset where the presence of the spurious feature entails the comment containing mentions of any of these categories: male, female, LGBT, black, white, Christian, Muslim, other religion. The presence of this spurious feature is correlated with the label "toxic".

**MultiNLI (MNL)** [35, 20]- is also a text classification task where a pair of sentences belongs to one of the three classes: Negation, Entailment, and Neutral. Spurious feature is the presence of negation words such as "no" or "never" and it is spuriously correlated with the Negation class.

**Celeb-A hair (Cel-A)** [33]- We select to implement the binary classification on the hair-color attribute to "Blond" and "non-Blond". The gender of the person is claimed to be the spurious feature. The correlation is between Female gender and being blonde.

| Dataset | #Classes | #Attributes | MaxGroup | MinGroup | Bias Level |
|---|---|---|---|---|---|
| cMNIST [29] | 10 | 10 | 5890 (10.71%) | 13 (0.02%) | 10.38 |
| Waterbirds [20] | 2 | 2 | 3498 (72.95%) | 56 (1.17%) | 3.67 |
| Urbancars-C [30] | 2 | 2 | 3800 (47.50%) | 200 (2.50%) | 1.90 |
| Metashift [31] | 2 | 2 | 789 (34.67%) | 196 (8.61%) | 1.41 |
| Civilcomments [34] | 2 | 2 | 148186 (55.08%) | 12731 (4.73%) | 1.45 |
| Nico-spurious [32, 62] | 6 | 6 | 3030 (32.53%) | 6 (0.06%) | 11.06 |
| Urbancars-B [30] | 2 | 2 | 3800 (47.50%) | 200 (2.50%) | 1.90 |
| Urbancars-A [30] | 2 | 4 | 3610 (45.12%) | 10 (0.12%) | 1.99 |
| MultiNLI [20, 35] | 3 | 2 | 67376 (32.68%) | 1521 (0.74%) | 2.28 |
| Celeb-A hair [33] | 2 | 2 | 71629 (44.01%) | 1387 (0.85%) | 1.62 |

Table 2: **Dataset statistics including the number of classes, attributes, the largest/smallest subgroups, and bias levels.**

# B   Experiment settings, models, and hyperparameters

## B.1   Class balancing

Prior work has shown that sample importance scores when used directly as a coreset selection can cause unintended class imbalances in the resulting coreset [57]. Another work [13] also perform a form of class balancing to ensure that none of the underrepresented classes are completely excluded from the selected subset. Since our experiments involve very small coreset sizes, and since the class labels are readily available, we implement a uniform class balancing strategy.

However, it should be noted that the datasets originally have imbalanced class distributions. Therefore, we calculate the ideal number of samples that each class should represent for a desired coreset size (equal proportions from all available classes), then if a particular class does not have enough samples, we iteratively divide the shortfall among the remaining classes until a distribution as close as possible to uniform is obtained.

## B.2   Baselines

Once the class-balancing has been applied and the number of samples to be picked from each class is calculated, the Random (R) selection policy uses uniform random selection on separate classes.

Random-groupbalanced (R-Gbal) baseline is implemented as an oracle baseline since it utilizes the group labels of each samples, that we assume we do not have in the current setting. First, based on the class-balancing constraint, we calculate the number of samples that should/could be sampled from each class. Then, within each class, we calculate the number of samples from each group that can be sampled such that the group distribution within each class is as close to uniform as possible. If a group does not have enough samples to create a uniform distribution, the shortfall is iteratively divided equally among the remaining groups until they run out of samples. This way, for a given size of coreset, the R-Gbal baseline selects the most group-balanced coreset possible without repeating the same samples (oversampling).

## B.3   Training surrogate model

**For datasets Waterbirds, Urbancars, Metashift, Nicospurious, Celeb-A hair**, we trained a ResNet50 [63] initialized with pretrained weights from Imagenet to calculate the sample-level scores for the learning-based selection methods. Following the setting proposed by [46], we trained the models with SGD with a constant learning rate of 0.001, momentum of 0.9, batch size 32 and a weight decay of 0.01. Following the previous work [27], for EL2N and Uncertainty, we trained the model for 20 epochs, and for Forgetting, we trained for 200 epochs

**For cMNIST**, we trained a ResNet18 [63] initialized with pretrained weights from Imagenet to calculate the sample-level scores for the learning-based selection methods. We trained the models

with SGD with a constant learning rate of 0.001, momentum of 0.9, batch size 32 and a weight decay of 0.01. For EL2N and Uncertainty, we trained the model for 20 epochs, and for Forgetting, we trained for 200 epochs

**For Civilcomments, MultiNLI**, we trained a pretrained Bert [64] model with Adam with learning rate 1e-5 and momentum 0.9 to calculate the sample-level scores for the learning-based selection methods. For EL2N and Uncertainty, we trained the model for 5 epochs, and for Forgetting, we trained for 20 epochs

For embedding-based methods: **SelfSup** and **SupProto**, we used the same model and pretrained weights as above, but did not train the feature extractor on the specific dataset; instead we extract the features for each sample from the penultimate layer. For the fine-tuned versions of the embedding-based scores: **SelfSup (finetuned)** and **SupProto (finetuned)** were first fine-tuned with supervision using the same training setting as EL2N, and the features are then extracted from the penultimate layer.

## B.4 Training downstream model

The same training recipe and models as the surrogate model were used here, except for the number of epochs trained. Since we compare models trained on a variety of coreset sizes, we keep the number of training iterations for each model constant. We train each model for a specific number of epochs such that the total number of iterations is equal to the number of iterations had the model been trained on the complete dataset for $n$ epochs. (Eg: for a coreset of size $2\%$, and $n$ is 100, the scaled number of training epochs would be $100/0.02 \simeq 5000$). We set $n$ for each dataset as follows: $n =100$ for **cMNIST, Waterbirds, Urbancars, Metashift, Nicospurious**, $n =50$ for **Celeb-A h**, and $n =10$ for **Civilcomments, MultiNLI**.

All models were trained with standard ERM with Stochastic Gradient Descent. All individual trainings were done on RTX-3090 GPUs with 24GB of VRAM. Total estimated compute for all experiments of this work is around 7,500 GPU hours.

# C Extended results

## C.1 Embedding-based characterization scores run a lower risk of inadvertently exacerbating bias compared to learning-based characterizations

Here we present the extended results corresponding to Section 4.1 from the main paper. Figure 6 contains the average precision evaluation of each characterization score on each dataset, when evaluated as a predictor for detecting bias conflicting samples. The random baseline is calculated by randomly ordering all the samples and then thresholding them at each level to calculate average precision, whereas the error bars represent the standard deviation. Therefore the average precision on random selection represents the overall proportion of bias-conflicting samples in the dataset. We see that across all datasets, leaning-based characterizations capture a much stronger signal that distinguishes bias conflicting samples from bias-aligning samples. We stipulate that this strong correlation between the characterization score and the bias-alignment of the samples can in turn cause inadvertent bias exacerbation when used as a metric for data selection. On the more challenging real-world datasets (Urbancars [30], Metashift [31], and Civilcomments [34], and Nico-spurious [32]), embedding-based methods do not appear to order the samples according to their bias levels (i.e., have near-random AP); even finetuning these embeddings (depicted by the shaded bars) does not significantly change these findings. It is also noteworthy that for datasets with more natural and complex spurious features (Urbancars-all [32], CelebAhair [33], and MultiNLI [35, 20], learning-based and embedding-based characterizations seem to capture signals of comparable strength.

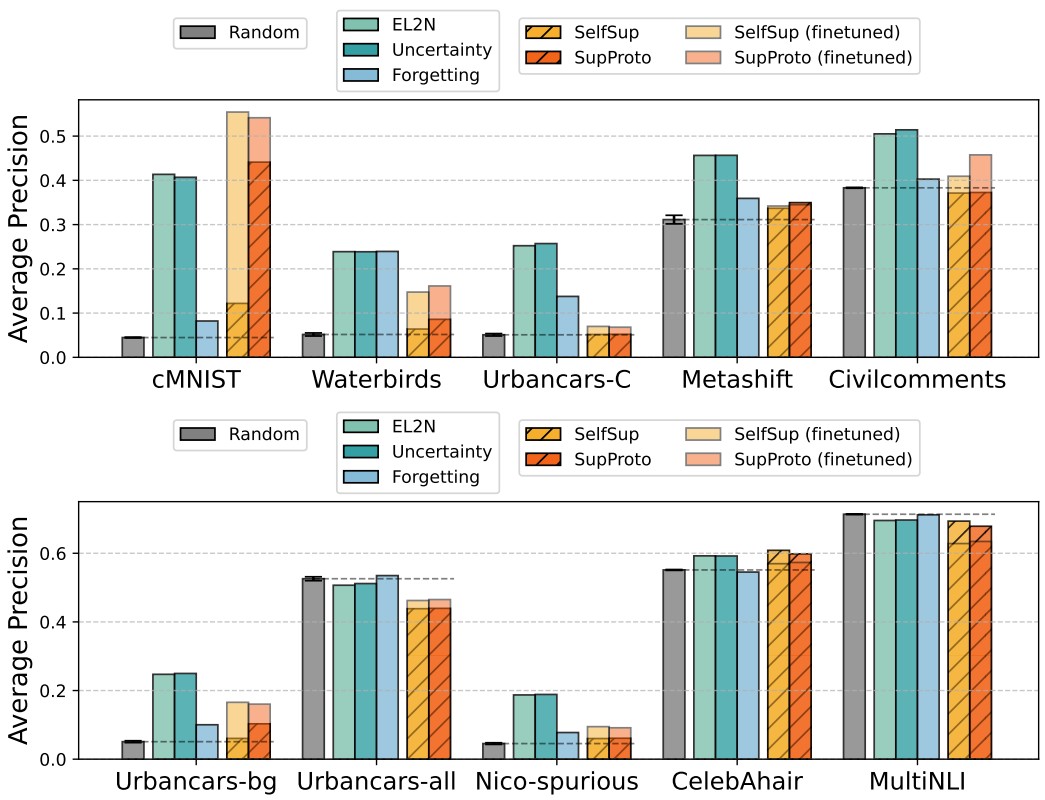

Figure 6: **Classifying bias-conflicting samples using characterization scores.** We measure the Average Precision of three learning-based methods (EL2N [27], Uncertainty [14] and Forgetting [15]) and two embedding-based methods (SelfSup [13] and SupProto [12]) at classifying bias-conflicting vs bias-aligning samples. The shaded bars on the embedding-based methods represent the results for scores generated from fine-tuned embeddings.

## C.2 Coreset bias level is not a consistent indicator of downstream robustness

In this section we include the extended results corresponding to Section 4.2 of the main paper. Figure 7 shows bias-levels, worst-group accuracy, and average accuracy for Difficult and Easy selection policies using EL2N [55] scores along with the baselines. The observations we outlined in the main paper consistently appear across all the datasets of the analysis. Coresets selected using the Difficult policies consistently have lower level of bias compared to those selected using the Easy policy. However, in the middle column we see that it does not always lead to improved robustness: **the Difficult coresets lead to more robust classifiers only when the coreset size is "sufficiently large."** What constitutes "sufficiently large" appears to further vary empirically between the 10 datasets used in this analysis. Furthermore, in the small data regime, the robustness of models for Difficult coresets becomes unintuitively low, despite the bias levels being the lowest out of all policies.

Corresponding numerical results (along with Median and Stratified selection policies) are shown in Table 3 and Table 4 respectively for moderate coreset sizes and very small coreset sizes. In the moderately sized coresets (40% and 60%), Difficult selections of EL2N scores yield high robustness, however this pattern is not consistent in the small coresets of 10% and 5%.

Extended results for all scoring methods and all selection policies for 40% selection rate is shown in Table 5

| Dataset | Baselines | | EL2N [55] scores | | | | SelfSup [13] scores | | | |
| --- | --- | --- | --- | --- | --- | --- | --- | --- | --- | --- |
| | R | R-Gbal | Diff | Strat | Med | Eas | Diff | Strat | Med | Eas |
| WB | 58.2 (96.6) | 83.3 (97.7) | **74.6** (98.0) | 73.1 (97.3) | 50.5 (96.4) | 30.2 (95.2) | 50.9 (96.1) | 51.0 (96.4) | 69.2 (97.1) | 37.1 (90.2) |
| c-Mn | 62.0 (99.8) | 84.4 (99.9) | **87.2** (99.9) | 74.0 (99.9) | 0.0 (99.0) | 0.0 (98.4) | 83.0 (99.7) | 83.5 (99.8) | 44.0 (99.7) | 0.0 (98.7) |
| CC | 77.4 (88.8) | 72.0 (89.6) | 63.0 (84.3) | 51.7 (91.8) | 63.9 (80.5) | 43.2 (68.1) | 77.8 (87.7) | 78.4 (88.3) | 70.3 (90.4) | **79.3** (86.0) |
| MSh | 73.8 (88.3) | 78.5 (87.2) | **78.5** (90.6) | 63.1 (89.3) | 66.5 (88.5) | 58.6 (86.4) | 73.3 (88.3) | 69.2 (90.3) | 74.3 (89.1) | 56.9 (87.5) |
| Nic-S | 44.0 (93.9) | 40.0 (94.9) | **44.0** (95.8) | 44.0 (95.3) | 34.0 (95.1) | 32.0 (93.7) | 44.0 (95.0) | 16.0 (93.3) | 34.0 (93.4) | 36.0 (94.3) |
| UC-C | 52.0 (86.8) | 61.6 (83.7) | 54.0 (89.2) | 51.2 (87.6) | 46.4 (70.3) | 49.6 (63.7) | 42.0 (86.7) | 48.4 (88.4) | **54.4** (85.0) | 44.0 (81.7) |
| UC-B | 44.8 (90.2) | 64.0 (87.6) | **52.0** (90.1) | 44.8 (89.5) | 26.4 (90.4) | 15.6 (91.8) | 43.2 (87.9) | 48.0 (88.8) | 41.6 (90.6) | 26.4 (89.4) |
| UC-A | 16.8 (96.4) | 50.4 (96.4) | **23.2** (97.2) | 19.2 (97.0) | 6.4 (94.9) | 7.2 (94.4) | 11.2 (96.5) | 17.6 (97.0) | 20.8 (96.6) | 9.6 (96.5) |
| MNL | 60.5 (78.8) | 57.6 (78.6) | 46.0 (61.5) | **65.6** (74.7) | 61.8 (79.0) | 46.0 (78.1) | 55.2 (79.4) | 60.8 (79.6) | 55.7 (78.2) | 58.3 (78.8) |
| Cel-A | 58.9 (93.4) | 71.1 (92.6) | 38.3 (94.9) | 47.8 (95.1) | 68.9 (93.2) | **79.0** (88.0) | 41.7 (94.4) | 66.1 (92.6) | 63.3 (94.4) | 78.9 (88.8) |

(a) 40 percent

| Dataset | Baselines | | EL2N [55] scores | | | | SelfSup [13] scores | | | |
| --- | --- | --- | --- | --- | --- | --- | --- | --- | --- | --- |
| | R | R-Gbal | Diff | Strat | Med | Eas | Diff | Strat | Med | Eas |
| WB | 67.3 (97.3) | 77.6 (98.0) | 73.2 (97.9) | 75.6 (97.4) | 28.6 (95.3) | 26.5 (94.9) | 73.7 (97.6) | 64.5 (97.1) | **75.9** (97.5) | 58.4 (94.8) |
| c-Mn | 78.9 (99.5) | 79.8 (99.5) | 63.1 (99.5) | 57.3 (99.6) | 1.0 (99.1) | 0.0 (99.0) | 78.0 (99.7) | **81.3** (99.8) | 65.0 (99.7) | 0.8 (99.4) |
| CC | 72.4 (90.1) | 70.8 (89.7) | 55.4 (92.1) | 53.7 (92.0) | 55.5 (75.7) | 57.3 (76.8) | 69.4 (90.0) | **71.2** (90.0) | 69.7 (90.5) | 61.6 (91.1) |
| MSh | 75.4 (90.7) | 75.4 (90.0) | 70.8 (90.7) | 70.8 (90.2) | 70.7 (89.1) | 67.5 (88.5) | 75.9 (90.7) | 75.4 (90.8) | **76.4** (89.7) | 61.5 (89.1) |
| Nic-S | 42.0 (94.7) | 40.0 (95.2) | 40.0 (96.2) | 38.0 (95.3) | 34.0 (95.6) | 32.0 (94.9) | 40.0 (96.3) | 34.0 (95.4) | 34.0 (95.0) | **44.0** (94.9) |
| UC-C | 49.2 (87.5) | 56.4 (85.2) | **54.4** (89.3) | 53.2 (89.3) | 49.6 (74.9) | 49.2 (73.7) | 48.4 (87.7) | 52.4 (90.1) | 52.8 (88.6) | 50.8 (85.4) |
| UC-B | 48.8 (88.7) | 58.0 (88.0) | **53.6** (90.7) | 49.6 (89.7) | 26.0 (90.9) | 24.4 (91.0) | 47.6 (89.9) | 49.2 (89.4) | 49.6 (90.4) | 39.2 (90.9) |
| UC-A | 22.4 (97.2) | 32.8 (97.5) | **23.2** (97.3) | 21.6 (97.5) | 10.4 (96.0) | 9.6 (95.9) | 16.8 (97.1) | 20.8 (96.9) | 20.8 (97.2) | 18.4 (97.3) |
| MNL | 58.0 (80.1) | 65.5 (80.5) | 65.3 (73.9) | 65.0 (79.6) | 55.2 (80.4) | 54.5 (80.4) | **67.2** (79.6) | 63.7 (80.7) | 65.9 (79.4) | 66.0 (79.5) |
| Cel-A | 73.3 (93.9) | 56.7 (95.3) | 61.1 (95.2) | 46.1 (95.2) | 68.9 (94.3) | **71.7** (93.0) | 63.3 (94.1) | 37.2 (95.8) | 55.6 (95.4) | 58.9 (94.5) |

(b) 60 percent

Table 3: **Worst-group accuracies and (Average accuracies) for different selection policies. For moderate coreset sizes: 40% and 60%.** The highest values of worst-group-accuracies are bolded, with second highest values underlined. The least robust, indicated by the least value for worst-group accuracy is shaded in brown. In general, Difficult selection policies with EL2N scores yield robust classifiers.

**(a) 5 percent**

| Dataset | Baselines | | EL2N [55] scores | | | | SelfSup [13] scores | | | |
|---|---|---|---|---|---|---|---|---|---|---|
| | R | R-Gbal | Diff | Strat | Med | Eas | Diff | Strat | Med | Eas |
| WB | 44.7 (95.3) | 89.0 (92.2) | 31.9 (85.2) | **51.1** (95.5) | 42.7 (95.3) | 24.8 (94.8) | 35.1 (94.9) | 26.3 (94.2) | 41.3 (95.0) | 29.1 (93.9) |
| c-Mn | 0.0 (99.2) | 95.3 (99.2) | **88.6** (99.9) | 8.0 (99.4) | 0.0 (97.1) | 0.0 (96.9) | 36.9 (99.5) | 24.7 (99.4) | 0.0 (98.2) | 0.0 (96.5) |
| CC | 66.6 (77.3) | 75.0 (81.2) | 7.2 (11.1) | 15.9 (19.1) | 58.0 (91.0) | 29.6 (56.0) | 42.4 (62.6) | 62.5 (75.9) | 68.3 (78.9) | **70.7** (80.8) |
| MSh | 55.4 (85.4) | 75.4 (87.5) | 6.2 (33.0) | 21.2 (51.2) | **56.9** (85.4) | 50.3 (83.0) | 55.0 (85.1) | 46.2 (84.8) | 52.3 (86.4) | 38.5 (83.5) |
| Nic-S | 34.0 (90.6) | 60.0 (88.8) | 34.0 (90.2) | **38.0** (89.9) | 28.0 (93.1) | 8.0 (88.7) | 26.0 (92.3) | 32.0 (93.0) | 30.0 (90.0) | 16.0 (82.2) |
| UC-C | 44.4 (75.5) | 52.8 (60.5) | 27.2 (59.7) | 44.8 (76.0) | 47.6 (66.8) | 46.0 (58.7) | 21.2 (67.3) | 40.8 (75.8) | **48.0** (70.7) | 38.8 (74.6) |
| UC-B | 21.6 (89.3) | 72.0 (78.1) | 14.0 (70.3) | 28.4 (85.1) | 17.2 (90.0) | 9.2 (90.4) | 9.2 (75.2) | **34.4** (84.6) | 20.4 (89.9) | 15.6 (86.0) |
| UC-A | 6.4 (95.1) | 62.4 (88.8) | 11.2 (76.3) | **15.2** (93.2) | 6.4 (94.5) | 3.2 (92.5) | 0.8 (82.3) | 8.8 (93.2) | 9.6 (95.5) | 4.8 (95.2) |
| MNL | 32.5 (69.4) | 62.3 (69.4) | 3.2 (17.5) | 22.2 (30.7) | 41.7 (74.9) | 37.7 (64.4) | 25.1 (60.1) | 34.3 (68.4) | **51.2** (71.1) | 29.7 (70.5) |
| Cel-A | 47.2 (94.8) | 83.3 (92.0) | 40.0 (82.1) | 51.1 (89.2) | 41.1 (93.6) | 80.6 (88.0) | **81.4** (86.7) | 64.4 (94.0) | 62.2 (93.2) | 80.9 (85.3) |

(a) 5 percent

| Dataset | Baselines | | EL2N [55] scores | | | | SelfSup [13] scores | | | |
|---|---|---|---|---|---|---|---|---|---|---|
| | R | R-Gbal | Diff | Strat | Med | Eas | Diff | Strat | Med | Eas |
| WB | 40.5 (95.3) | 88.6 (95.6) | **68.5** (96.7) | 67.4 (96.3) | 47.2 (95.9) | 29.1 (95.1) | 40.8 (94.8) | 34.7 (95.3) | 55.9 (96.0) | 23.9 (92.4) |
| c-Mn | 24.8 (99.4) | 95.0 (99.5) | **87.9** (99.9) | 47.0 (99.8) | 0.0 (98.1) | 0.0 (97.3) | 69.4 (99.6) | 73.4 (99.6) | 0.0 (98.8) | 0.0 (96.5) |
| CC | 77.9 (85.3) | 72.2 (80.3) | 5.0 (10.0) | 18.7 (21.9) | **80.6** (88.4) | 34.8 (58.1) | 57.0 (73.4) | 56.9 (79.8) | 68.3 (79.8) | 68.4 (79.3) |
| MSh | 58.5 (88.4) | 81.7 (88.8) | 11.4 (42.9) | 63.1 (82.2) | **67.5** (87.3) | 54.5 (84.7) | 57.1 (85.4) | 56.9 (87.7) | 58.5 (86.8) | 38.5 (84.7) |
| Nic-S | 32.0 (94.2) | 40.0 (81.2) | 36.0 (95.4) | **50.0** (92.3) | 30.0 (94.0) | 16.0 (91.2) | 30.0 (94.5) | 26.0 (93.5) | 26.0 (91.9) | 28.0 (88.3) |
| UC-C | 48.8 (75.4) | 60.0 (67.0) | **53.6** (83.0) | 46.0 (80.0) | 46.4 (65.9) | 47.6 (57.4) | 27.6 (73.0) | 46.4 (81.8) | 48.8 (74.8) | 40.4 (75.9) |
| UC-B | 32.4 (88.8) | 76.4 (82.1) | **44.4** (86.4) | 28.0 (88.1) | 21.6 (90.6) | 11.2 (90.7) | 17.6 (80.6) | 26.0 (88.1) | 26.0 (89.9) | 17.6 (88.1) |
| UC-A | 8.8 (95.7) | 64.8 (92.7) | **27.2** (94.3) | 13.6 (94.9) | 5.6 (94.4) | 4.8 (92.4) | 3.2 (88.9) | 11.2 (96.4) | 11.2 (95.5) | 6.4 (94.8) |
| MNL | 58.4 (73.6) | 63.2 (73.2) | 7.7 (19.5) | 12.7 (27.6) | 46.5 (77.8) | 45.6 (70.6) | 38.6 (67.7) | **48.6** (73.7) | 44.9 (74.1) | 43.6 (74.2) |
| Cel-A | 75.6 (92.0) | 76.1 (92.0) | 41.7 (90.1) | 63.3 (91.8) | 72.8 (92.6) | 70.2 (79.4) | **77.2** (84.8) | 76.7 (91.4) | 48.3 (92.1) | 76.6 (82.1) |

(b) 10 percent

Table 4: **Worst-group accuracies and (Average accuracies) for different selection policies. For very small coreset sizes: 5% and 10%.** The highest values of worst-group-accuracies are bolded, with second highest values underlined. The least robust, indicated by the least value for worst-group accuracy is shaded in brown. Difficult selection suffers from a large drop in both average and worst-group accuracies, especially in EL2N. Selection policies that incorporate less difficult samples tend to yield comparatively higher robustness.

| Dataset | Baselines | | EL2N [27] | | | | Uncertainty [14] | | | | Forgetting [15] | | | | SelfSup [13] | | | | SupProto [12] | | | |
|---|---|---|---|---|---|---|---|---|---|---|---|---|---|---|---|---|---|---|---|---|---|---|
| | R | R-gbal | Diff | Strat | Med | Eas | Diff | Strat | Med | Eas | Diff | Strat | Med | Eas | Diff | Strat | Med | Eas | Diff | Strat | Med | Eas |
| C-Mn | 62.0 | 84.4 | **87.72** | 74.0 | 0.0 | 0.0 | 37.8 | **82.3** | 1.0 | 0.0 | **9.9** | 4.0 | 0.0 | 0.0 | 83.0 | **83.5** | 44.0 | 0.0 | **87.0** | 48.5 | 0.0 | 0.0 |
| WB | 58.2 | 83.3 | **74.6** | 73.1 | 50.5 | 30.2 | **72.6** | 71.2 | 51.0 | 28.2 | 75.4 | **79.6** | 53.8 | 52.0 | 50.9 | 51.0 | **69.2** | 37.1 | **66.5** | 58.8 | 59.4 | 30.0 |
| UC-c | 52.0 | 61.6 | **54.0** | 51.2 | 46.4 | 49.6 | 54.8 | **55.2** | 46.4 | 50.4 | 51.2 | **54** | 31.6 | 31.6 | 42.0 | 48.4 | **54.4** | 44.0 | 46.8 | **50.4** | 50.0 | 44.8 |
| MSh | 73.8 | 78.5 | **78.5** | 63.1 | 66.5 | 58.6 | **73.8** | 69.2 | 70.2 | 65.4 | 71.7 | **76.4** | 46.6 | 48.7 | 73.3 | 69.2 | **74.3** | 56.9 | 67.0 | **73.8** | 72.3 | 58.5 |
| CC | 77.4 | 72.0 | 63.0 | 51.7 | **63.9** | 43.2 | 60.2 | 64.2 | 61.0 | **75.2** | 60.9 | 65.3 | **82.0** | 79.5 | 77.8 | 78.4 | 70.3 | **79.3** | 76.2 | **77.1** | 74.1 | 75.2 |
| Nic-s | 44.0 | 40.0 | **44.0** | **44.0** | 34.0 | 32.0 | **44.0** | 42.0 | 36.0 | 30.0 | 44.0 | 40.0 | **46.0** | 40.0 | **44.0** | 16.0 | 34.0 | 36.0 | 38.0 | **42.0** | **42.0** | 38.0 |
| UC-b | 44.8 | 64.0 | **52.0** | 44.8 | 26.4 | 15.6 | 52.8 | **53.2** | 26.0 | 15.6 | 47.6 | 50.8 | **54.0** | **54.0** | 43.2 | **48.0** | 41.6 | 26.4 | **53.2** | 52.4 | 36.8 | 13.6 |
| UC-a | 16.8 | 50.4 | **23.2** | 19.2 | 6.4 | 7.2 | 22.4 | **26.4** | 8 | 7.2 | 20.0 | **23.2** | 16.0 | 16.0 | 11.2 | 17.6 | **20.8** | 9.6 | **20.8** | 19.2 | 17.6 | 5.6 |
| MNL | 60.5 | 57.6 | 46.0 | **65.6** | 61.8 | 46.0 | 61.5 | **64.0** | 53.8 | 49.2 | **59.5** | 57.8 | 52.4 | 52.4 | 55.2 | **60.8** | 55.7 | 58.3 | - | **63.9** | 62.6 | 55.4 |
| Cel-h | 58.9 | 71.1 | 38.3 | 47.8 | 68.9 | **79.0** | 42.2 | 42.2 | 62.8 | **78.9** | 27.2 | 42.8 | **64.4** | **64.4** | 41.7 | 66.1 | 63.3 | **78.9** | 50.0 | 59.4 | 63.9 | **82.2** |

Table 5: **Worst-group accuracies of different selection policies within each scoring methods (learning-based: EL2N [27], Uncertainty [14], Forgetting [15], and embedding-based: SelfSup [13], SupProto [12]) at 40% selection rate.** The highest worst-group-accuracies within each scoring method for each dataset are bolded.

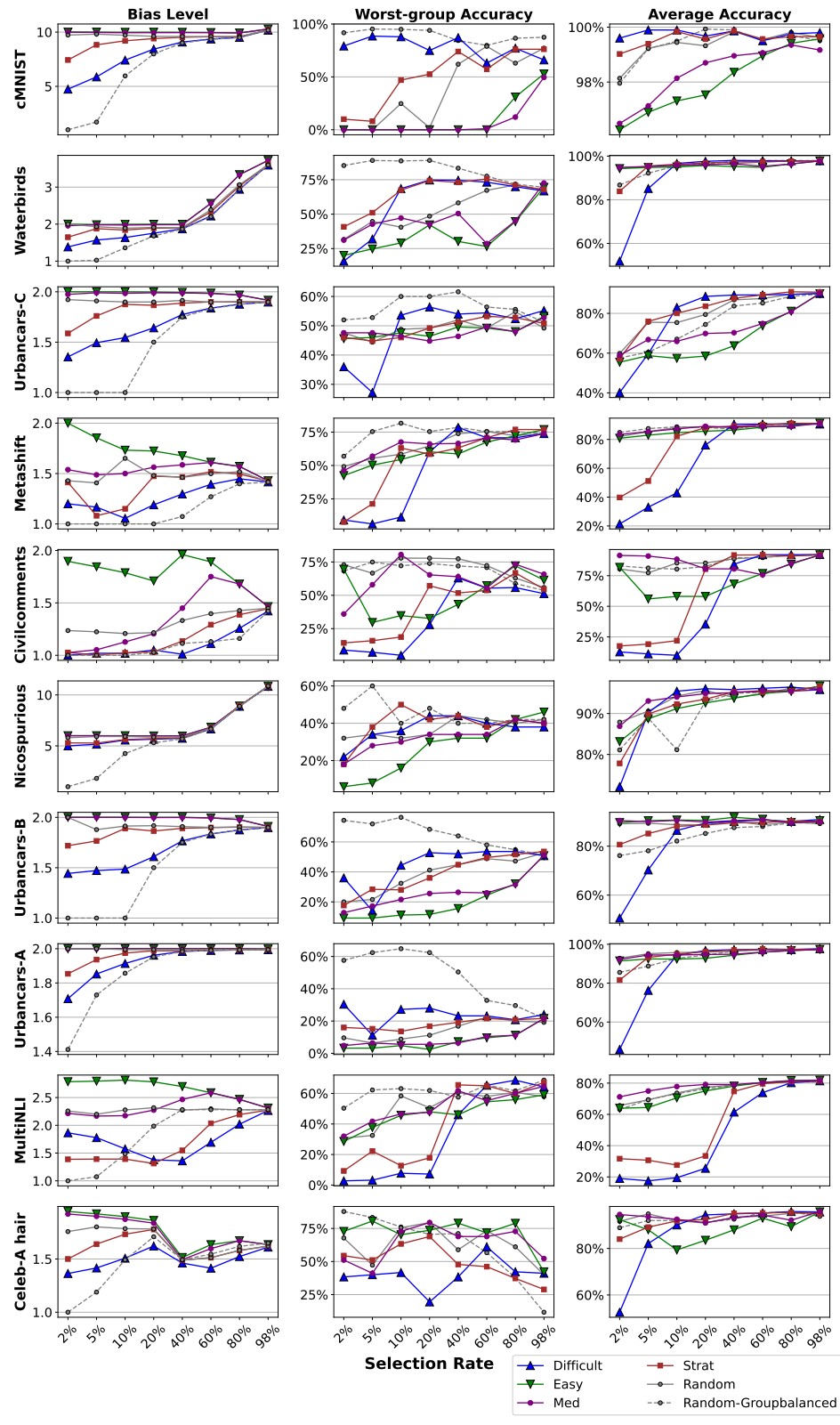

Figure 7: **Data bias and classifier accuracies for different selection policies using EL2N scores.** Selecting the Difficult samples typically results in less biased coresets and corresponding more robust (highest worst-group accuracy) classifiers than Easy samples at higher selection rates. The Difficult samples also lead to more robust models than Random selection. However, as coreset size gets smaller, we see a significant drop in average and worst-group accuracies for Difficult samples. In such settings, Stratified and Median policies, which are consistently more biased than Difficult, counter-intuitively yield higher robustness.

## C.3 Trading off most difficult bias-conflicting samples to improve robustness

This section includes the extended results corresponding to Section 4.3 of the main paper. Difficult* selection policy is a simple heuristic where a small percentage (3%) of the highest scoring samples is removed from Difficult selection. Bias levels, worst-group accuracy, and average accuracy for this heuristic policy along with the rest of the policies are applied on EL2N scores for all the datasets of the analysis as shown in Figure 8. We can see that all methods: Difficult*, Stratified, and Median make the coresets progressively more biased compared to Difficult selection. However, they result in improved robustness (worst-group accuracy) in cases where Difficult policy has catastrophically low robustness.

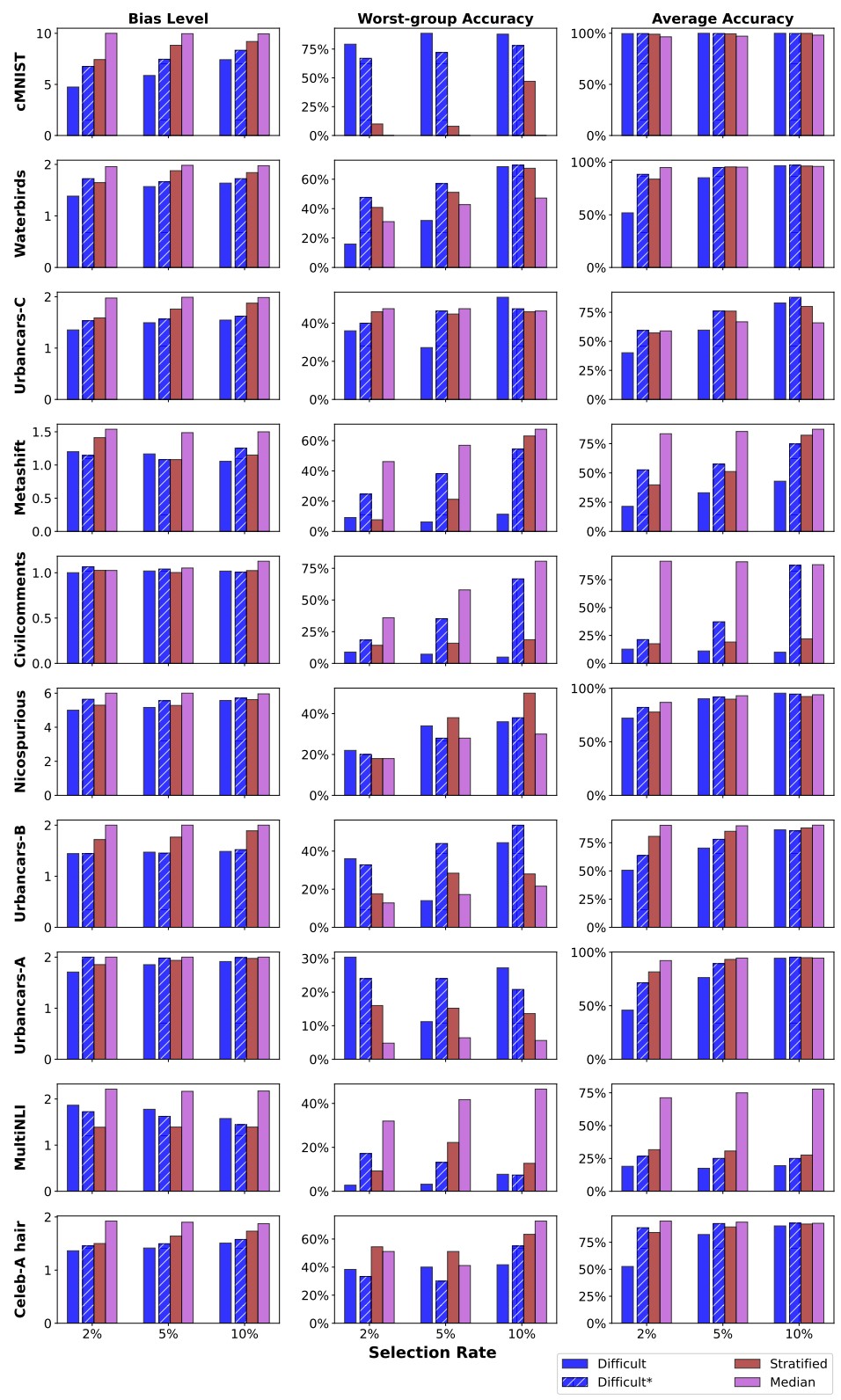

Figure 8: **Trading off most difficult minority samples to achieve higher robustness.** Difficult\*, Median and Stratified selection policies often make the selected coreset slightly biased, however, they improve the robustness of the downstream models, in settings where Difficult selection has a catastrophic drop in performance.

## C.4 Further exploration into catastrophic loss of accuracy for difficult coresets in small data regime

We inspect train-test accuracy gaps for models trained on small-data regime coresets and observe that all models achieve 100% training accuracy. The drop in average accuracy for difficult coresets in the small data regime therefore suggests a high level of overfitting compared to other policies, also consistent with claims of Sorscher et al [13] as we discussed in Section 4.2.

It has been hypothesized that the catastrophic drop in accuracy for difficult coresets could be due to the coreset achieving low coverage on the data space. Zhen et al. [11] utilized the concept of p-partial r-cover to quantify this phenomenon, where r is some radius around each data of the coreset and p is the proportion of training data covered. We used their p-partial r-cover as a metric to investigate whether this pattern persists in datasets with strong spurious correlations. Using the features of ImageNet pre-trained ResNet-50, we selected a radius r where it would cover 95% of the training data. Using the r, we obtained the measured p from the selected coresets shown in Table 6. Significant decrease in p in difficult selected coresets at 5% and 2% across the datasets confirms the drop in coverage.

| Selection Rate | Random (baseline) | Difficult | Easy |
|---|---|---|---|
| 2% | 80.3 | **45.4** | 82.3 |
| 5% | 85.7 | **58.8** | 86.1 |
| 20% | 93.3 | 92.4 | 90.9 |

(a) Waterbirds [20]

| Selection Rate | Random (baseline) | Difficult | Easy |
|---|---|---|---|
| 2% | 61.0 | **7.9** | 71.1 |
| 5% | 74.7 | **25.0** | 77.4 |
| 20% | 89.6 | 80.0 | 88.4 |

(b) Metashift [31]

Table 6: **p-partial r-cover achieved by Difficult and Easy selection policies for Waterbirds [20] and Metashift [31] datasets.** At low selection rates, Difficult selection yeilds to significantly less coverage than Easy of Random selections (bolded).

