# The Impact of Coreset Selection on Spurious Correlations and Group Robustness

**Amaya Dharmasiri**[1]    **William Yang**[1]    **Polina Kirichenko**[2]

**Lydia T. Liu**[1]    **Olga Russakovsky**[1]

[1]Princeton University    [2]FAIR at Meta

{amayadharmasiri, williamyang, ltliu, olgarus}@princeton.edu,
polkirichenko@meta.com

# Appendix

This appendix includes further details on the experiment setup and analysis of the paper "The Impact of Coreset Selection on Spurious Correlations and Group Robustness".

- A - Details of all the datasets used in our extensive analysis, including their bias levels
- B - Implementation details of the analysis pipeline, model details and hyperparameters