# OpenReview forum: "The Impact of Coreset Selection on Spurious Correlations and Group Robustness"
_NeurIPS.cc/2025/Datasets_and_Benchmarks_Track — NeurIPS 2025 Datasets and Benchmarks Track poster_

### Official Review · Reviewer_cikP · 2025-06-19

**Rating:** 4
**Confidence:** 2

**Summary:**

This paper delivers the first systematic study of how coreset selection—the practice of training on a strategically chosen data subset—interacts with hidden spurious correlations and downstream group robustness. Using ten vision- and NLP-benchmarks with known spurious features, the authors cross a grid of five sample-scoring heuristics (three learning-based, two embedding-based) and five selection policies over a wide range of subset sizes. They discover that learning-based scores like EL2N and Uncertainty can strongly magnify or suppress bias depending on whether one keeps the “Difficult’’ or “Easy’’ samples, whereas embedding-based scores are largely bias-agnostic. Selecting the highest-scoring (hardest) examples often lowers dataset bias and improves worst-group accuracy—but only when the subset exceeds a “sufficiently large’’ threshold; in the small-data regime it can back-fire, while overly “Easy’’ subsets achieve deceptively high average accuracy yet poor fairness. The study culminates in a practical heuristic: prefer difficult, rare samples and ensure the coreset remains big enough to avoid coverage collapse.

**Dataset Code Accessibility:**

No

**Ethical Considerations:**

No, there are no or only very minor ethics concerns

**Limitations Weaknesses:**

The analysis, while extensive, is confined to supervised classification tasks and to worst-group accuracy; it does not test other modalities (e.g., generative models), objectives (e.g., detection) or fairness metrics, so generalisability is uncertain. No new coreset algorithm is proposed; the study is diagnostic rather than prescriptive, and its heuristic recommendations lack theoretical guarantees. All experiments rely on publicly labelled datasets whose spurious attributes are already documented; true “unknown’’ biases in the wild may be harder to expose. Finally, large-scale sweeps require significant compute, and the conclusions (e.g., the exact “sufficiently large’’ threshold) sometimes vary by dataset, which could limit plug-and-play adoption.

**Strengths Contributions:**

A major strength is the breadth and rigor of the experimental matrix: ten diverse datasets, five scoring metrics and five policies yield hundreds of controlled runs, exposing nuanced, previously overlooked interactions between sample difficulty, bias alignment and robustness. The work goes beyond reporting averages, consistently analyzing worst-group accuracy and an explicit bias-level statistic, thus foregrounding fairness considerations. It overturns two common folk-theories—that low-bias subsets always help robustness, and that “hard example mining’’ is uniformly beneficial—by showing their conditional validity. The authors also identify safer default choices (embedding scores; stratified or median policies) and offer a simple “Difficult*’’ tweak that drops a few extreme outliers to rescue small-subset performance, providing actionable guidance for data-centric ML practitioners.

---

> ### Author Rebuttal · Authors · 2025-07-30
>
> Thank you for your thoughtful and encouraging review. We're especially glad that you highlighted the **breadth and rigor of the experimental matrix**, which enabled us to **expose nuanced, previously overlooked interactions** of various concepts in this space. We also appreciate you noticing our focus on **worst-group accuracy and an explicit bias-level statistic** to **foreground fairness considerations** as this was a central goal of our work. Your recognition that our work **overturns two common folk-theories and offers actionable guidance for data-centric ML practitioners** is deeply appreciated.
>
> >1. The analysis, while extensive, is confined to supervised classification tasks and to worst-group accuracy; it does not test other modalities (e.g., generative models), objectives (e.g., detection) or fairness metrics, so generalisability is uncertain.
>
> That is correct, our analysis is focused on the image classification task. As mentioned in the abstract, we “conduct the first comprehensive analysis of the implications of data selection on the bias levels of the selected coresets and the robustness of downstream models trained on them.” We do not claim generalizability beyond this setting.
>
> With regard to different fairness metrics, we considered alternative metrics like the variance across subgroup accuracies, but as those are directly related to the worst-group-accuracy metric that we used in our work, we chose not to further complicate the paper. We are not aware of orthogonal fairness metrics that could be meaningfully used in this setting of group-robustness to spurious correlations, but we would welcome suggestions. Furthermore, worst-group accuracy has been the standard metric to evaluate group robustness, as used by numerous seminal works in this domain. [20, 46, 48]
>
> [20] Sagawa et al. ICLR 2019
>
> [46] Kirichenko et al. ICLR 2023
>
> [48] Liu et al. ICML 2021
>
> >2. No new coreset algorithm is proposed; the study is diagnostic rather than prescriptive, and its heuristic recommendations lack theoretical guarantees.
>
> Yes, that is correct, we do not intend to propose a new coreset algorithm.
>
> In fact, one of the challenges is that some of our concerning findings seem to apply to all the coreset algorithms we tested and point to some of the (perhaps inevitable?) challenges with data reduction techniques.
>
> We do, however, aim to provide a more prescriptive set of suggestions for selecting coresets that ensure downstream group-robustness when group labels are not annotated in a dataset. (Please see response 2 to reviewer KrE1 above.)
>
> >3. All experiments rely on publicly labelled datasets whose spurious attributes are already documented; true “unknown’’ biases in the wild may be harder to expose.
>
> In this work, we have used datasets whose spurious attributes are documented and publicly available to evaluate group-level performance. **We want to clarify that none of the coreset selection methods implemented in the paper use spurious feature annotations to inform their coreset selection.**. Thereby, our experimental setting simulates how coreset selection methods interact with true “unknown” biases in the wild, as mentioned by the reviewer.
>
> We use the documented spurious attributes of these public datasets to identify and isolate different subgroups in the test distribution to detect and explore performance disparities across the subgroups (ie, evaluating accuracy within individual groups to measure worst-group accuracy). Additionally, as mentioned in the Experimental setup, we implemented “an oracle R-Gbal (Random-Groupbalanced) which utilizes group labels to create a random group-balanced coreset. R-Gbal acts as an upper bound since it utilizes the group information, which we assume is unavailable in our setting.”
>
> As established by Section 4.2, despite not having access to spurious attribute labels, some coreset scores carry a strong signal toward bias-conflicting samples. This implicit signal plays a core role in our analysis.
>
> We also reiterate that our prescriptive suggestions (please see response 2 to reviewer KrE1 above) are intended for practitioners who do not have access to spurious feature labels, and therefore cannot conduct elaborate preemptive robustness checks.
>
> >4. Finally, large-scale sweeps require significant compute, and the conclusions (e.g., the exact “sufficiently large’’ threshold) sometimes vary by dataset, which could limit plug-and-play adoption.
>
> **This comment is probably based on a misunderstanding of the intention of our analysis. We are not proposing an analysis pipeline that is supposed to be a plug-and-play.**
> In a real application scenario, implementing all the different types of selections and policies to select the best option is infeasible. The real challenge is when the spurious correlations are unknown and the dataset does not contain group labels to preemptively evaluate the robustness of each coreset option.
> Instead, we used our extensive analysis, which included large-scale sweeps, to unravel nuances and consistent patterns that are supposed to inform practitioners in their respective downstream coreset selection tasks.
>
> About the “sufficiently large” threshold, it does indeed vary by dataset. However, as elaborated in response 4 to reviewer KrE1, this threshold can be approximated using average accuracy. If the average accuracy reached by the same model trained on your difficult coreset is significantly lower than that of a random coreset of the same size, it signals that your difficult coreset is not “sufficiently large”. We argue that this simple diagnostic could be done by any practitioner.
>
> *Finally, we noticed that you have [by mistake?] indicated that the code is unavailable for this work*. **This is not correct, as we have made our code and analyses publicly available**, as mentioned in the openreview submission, as well as in the supplementary materials.

---

> > ### Comment · Reviewer_cikP · 2025-08-04
> >
> > The authors provide thoughtful rebuttals，I will maintain my score.

---

### Official Review · Reviewer_KrE1 · 2025-06-30

**Rating:** 4
**Confidence:** 3

**Summary:**

This paper presents the first comprehensive empirical investigation into the effects of coreset selection methods on spurious correlations and the group robustness of downstream models. The authors conduct a large-scale study spanning multiple datasets with known spurious biases, five sample characterization scores, and five selection policies across a wide range of coreset sizes. Their analysis reveals several non-trivial insights: (1) embedding-based sample characterization scores are less likely to amplify dataset bias than learning-based scores; (2) reducing bias in a coreset does not automatically lead to a more robust downstream model, particularly for small coreset sizes; and (3) selecting for easy, prototypical samples can maintain high average accuracy at very small coreset sizes while masking a catastrophic drop in worst-group robustness.

**Additional Feedback:**

1. As stated ‘What constitutes “sufficiently large” appears to vary empirically between datasets’ , the analysis is based on a single method. Will such a ‘turning point’ change when switching to a different method?

2. Why not include experiments that are directly trained on these datasets? Will using pretrained weights carry the bias inherent from ImageNet-1K.

3. How to explain the abnormal behavior for NLP dataset (i.e., Civilcomments) in Fig.3? The easy policy has a very high performance at 2%, even for the worst-group accuracy.

**Dataset Code Accessibility:**

Yes

**Dataset Code Comments:**

yes

**Ethical Considerations:**

No, there are no or only very minor ethics concerns

**Final Justification:**

Most of my concerns are addressed and I will raise my score.

**Limitations Weaknesses:**

1. Most of the experiments are empirical and lack theoretical analysis. The paper observes that Difficult coresets lead to a catastrophic performance drop in the small-data regime and hypothesizes this is due to low coverage or overfitting. This analysis could be strengthened with more direct, diagnostic experiments. Could the authors include a brief analysis to support these hypotheses? For example, one could visualize the feature space coverage of different coresets (e.g., using t-SNE) or measure the train-test accuracy gap for these models to provide more direct evidence of overfitting.

2. The paper's conclusion could be more prescriptive. It would be highly beneficial to add a small section or a summary table with practical guidelines. For instance: "If your primary goal is to avoid catastrophic drops in robustness with a very small coreset (<10%), our results suggest that Stratified or Median selection policies are safer choices than a pure Difficult policy.

3. The experiments rely on models pretrained on ImageNet, which is known to have its own biases. This is a potential confounding variable. What's the view of authors on this point?

**Strengths Contributions:**

1. The paper tackles a critical and underexplored intersection: the trade-offs between training efficiency (via coreset selection) and model fairness/robustness. By being the first to systematically chart this territory, the paper makes a foundational contribution.

2. The comprehensiveness of the experiments is the paper's greatest strength. The evaluation across ten datasets, multiple scoring paradigms, and a wide range of selection rates provides a very strong evidentiary basis for the claims. This "move-the-earth" effort ensures the findings are not just anecdotal but represent robust patterns.

3.  The paper excels at moving beyond surface-level observations to uncover non-trivial nuances. The distinction between score families (Sec 4.1), the decoupling of coreset bias from model robustness (Sec 4.2), and the specific analysis of the small-data regime (Sec 4.3) are all high-quality insights that challenge common assumptions in the community.

---

> ### Author Rebuttal · Authors · 2025-07-31
>
> Thank you for your detailed and generous review! We are happy to see you recognize our paper as **making a foundational contribution by addressing the critical and underexplored intersection of coreset selection and robustness trade-offs**, and **excels at moving beyond surface-level observations to uncover non-trivial nuances that challenge common assumptions in the community**.
>
> ### 1. **Overfitting and catastrophic performance drop in small data regime.**
> Thank you for this suggestion! We inspect train-test accuracy gaps for models trained on small-data regime coresets and observe that all models achieve 100% training accuracy. Therefore, the train-test accuracy gap corresponds to the test error reported in the paper (Figure 3, third column with “Average Accuracy”). The drop in average accuracy for difficult coresets in the small data regime indeed suggests a **high level of overfitting** compared to other policies, also consistent with claims of Sorscher et al [13] as we discussed in Section 4.2.
>
> In terms of coverage, we will utilize the concept of p-partial r-cover from Zhen et al [11] where r is some radius around each data of the coreset and p is the proportion of training data covered. Using the features of ImageNet pre-trained ResNet-50, we selected a radius r where it would cover 95% of the training data. Using the r, we obtained the measured p from the selected coresets shown in the table below:
>
> Waterbirds
> | Selection Rate | Random (baseline) | Difficult | Easy  |
> |----------------|-------------------|-----------|-------|
> | 2             | 80.3             | **45.4**     | 82.3 |
> | 5             | 85.7             | **58.8**     | 86.1 |
> | 20            | 93.3             | 92.4     | 90.9 |
>
> Metashift
> | Selection Rate | Random (baseline) | Difficult | Easy  |
> |----------------|-------------------|-----------|-------|
> | 2             | 61.0             | **7.9**      | 71.1 |
> | 5             | 74.7             | **25.0**     | 77.4 |
> | 20           | 89.6             | 80.0     | 88.4 |
>
> Significant decrease in p in difficult selected coresets at 5% and 2% across the datasets confirms the drop in coverage.
>
> ### 2. Prescriptive suggestions
> We appreciate this suggestion. The consistency of observed patterns enables prescriptive guidance for practitioners, which we will highlight in the camera-ready version:
>
> Given: (1) A training+validation set with class labels but no spurious annotations, (2) a desired coreset size range $[N_\text{min}, N_\text{max}]$ (which is dictated by the constraints of the application).
>
> Goal: Find the smallest coreset likely to yield high accuracy and group robustness on a test set with the same target classes and unknown group labels.
>
> #### Method:
>
> **Initialize:** Set coreset size $N = N_{\text{min}}$
>
> **While** $N \leq N_{\text{max}}$:
>
> 1. Select two coresets of size $N$:
> - $S_{N,\text{difficult}}$ using the **difficult** policy
> - $S_{N,\text{random}}$ using the **random** policy
>
> 2. Train models:
> - $M_{N,\text{difficult}}$ on $S_{N,\text{difficult}}$
> - $M_{N,\text{random}}$ on $S_{N,\text{random}}$
>
> 3. **If** $\text{avg accuracy}(M_{N,\text{difficult}}) \approx \text{avg accuracy}(M_{N,\text{random}})$:
>
> - **Return** $S_{N,\text{difficult}}$
>     → $N$ is “sufficiently large” and a difficult-sample coreset likely leads to high group robustness
>
> 4. **Else:**
>
> - Increase $N$
>     → Current $N$ falls in the “small-data regime” and $S_{N,\text{difficult}}$ may not ensure robustness
>
> **Fallback**:
> If $N_\text{max}$ is reached, use Stratified or Median policies. While not always optimal (_“these strategies are still far from being one-size-fits-all solutions”_ ), they are more reliable than difficult selection in the small-data regime.
>
> ### 3. Using models pretrained on ImageNet and fine-tuned on downstream tasks.
> Using ImageNet pretrained models is a standard setting in spurious correlation literature [e.g., see 20, 46, 48]. Most of the datasets with spurious feature annotations are limited in size (e.g., Waterbirds has a total of 4.8k training images), so training from scratch, even on the full dataset, leads to weak average performances (see [46] Appendix C). Data limitation relative to model size is even more pronounced in the coreset selection setting — 2% of the waterbirds dataset is only 96 images.
>
> Below is the performance on Waterbirds coresets using EL2N scores with ResNet-50 trained from scratch:
>
> | Selection ratio | R: bias level | R: WGA (Avg Acc) | Diff: bias level | Diff: WGA (Avg Acc) |
> |-----------------|--------------|------------------|------------------|----------------------|
> | 2%              | 1.91         | **4.0** (28.1)       | **1.44**         | 2.7 (26.3)           |
> | 10%             | 1.89         | **17.3** (82.8)  | **1.37**         | 8.7 (15.7)           |
> | 20%             | 1.89         | 15.1 (86.9)      | **1.71**         | **34.3** (40.5)      |
> | 40%             | 1.92         | 16.2 (85.0)      | **1.87**         | **24.6** (78.9)      |
> | 100%            |  3.67            |         12.61 (87.60)         |                  |                      |
>
> Comparing with the results from Figure 3, here we show that **our conclusions hold for ResNets trained from scratch**, ensuring that our results are not confounded by pretraining biases. Difficult selection still yields the most unbiased coresets. For coreset sizes smaller than 20%, Difficult selection doesn’t yield the most robust classifiers, even though it has the lowest bias levels.
>
> Worst-group-accuracy, even if we train on the whole dataset, is well below random chance. Therefore, this is not a suitable setting to uncover the subtleties of how classifiers trained on coresets interact with spurious correlations.
>
> ### 4. Turning point between “small” to “sufficiently large” data regime for different coreset selection methods.
>
> Our results show this varies slightly by method. However, a simple diagnostic works, regardless of the selection method: If the average accuracy reached by the same model trained on the difficult coreset is significantly lower than that of a random coreset of the same size, it signals that this difficult coreset is not “sufficiently large”. If the average accuracies are comparable, the difficult coreset is sufficiently large, and can lead to more robust classifiers.
>
> We present additional results for Difficult coresets from the other methods of the learning-based family. For each selection rate, we report worst group accuracy, with corresponding average accuracy within parentheses. The values beyond what we can consider “sufficiently large” have been bolded.
>
> Difficult coresets from Uncertainty and Forgetting both reach sufficiently large beyond 10% selection rate on **Waterbirds**:
> | Selection Rate     | 5%          | 10%         | 20%         | 40%         | 60%         |
> |---------------------|-------------|-------------|-------------|-------------|-------------|
> | Random              | 43.3 (95.1) | 45.5 (95.5) | 53.1 (96.3) | 62.0 (96.8) | 71.3 (97.4) |
> | Diff (Uncertainty)  | 37.7 (86.6) | **67.9 (96.5)** | **72.3 (97.7)** | **72.6 (97.9)** | **73.7 (98.0)** |
> | Diff (Forgetting)   | 23.8 (27.6) | **81.5 (91.7)** | **78.8 (96.0)** | **75.4 (96.8)** | **79.0 (97.6)** |
>
> Difficult coresets from Uncertainty are “sufficiently large” beyond 40%, and Forgetting beyond 20% selection rates on **Metashit**:
> | Selection Rate        | 5%         | 10%         | 20%         | 40%         | 60%         |
> |-----------------------|------------|-------------|-------------|-------------|-------------|
> | Random                | 62.0 (86.6)| 61.5 (88.6) | 66.9 (89.4) | 73.8 (89.2) | 74.6 (90.9) |
> | Diff (Uncertainty)    | 6.2 (33.0) | 11.4 (43.1) | 60.7 (76.4) | **73.8 (91.2)** | **70.8 (90.7)** |
> | Diff (Forgetting)     | 5.9 (24.7) | 51.6 (67.7) | **75.1 (83.7)** | **71.7 (88.4)** | **75.4 (90.6)** |
>
> In both, average accuracy gap between difficult vs. random selections reliably indicates coreset size sufficiency.
>
> ### 5. WGA trend on CivilComments for Easy selection policy.
> We hypothesize that the abnormal behavior at 2% selection rate is an artifact of strong class imbalance of the original dataset, and the Easy policy’s strong preference to select bias-aligning samples. The classification task is between two classes: Toxic and Neutral comments. Toxic comments are spuriously correlated with the presence of a trigger word, and Neutral comments with the absence of it (Please see Supplementary). As explained in Section 3: Experimental setup, during coreset selection we apply class-balancing to avoid any additional biases.
>
> Original group proportions of the dataset show a significant class imbalance.
> |                | Trigger word | No Trigger word | Total |
> |----------------|--------------|------------------|--------|
> | **Toxic**      | 0.07         | 0.05             | **0.12**   |
> | **Neutral**    | 0.33         | 0.55             | **0.88**   |
> | **Total**      | 0.40         | 0.60             | 1.00   |
>
> Proportion of each group -Easy Selection at 2% (classes are balanced during selection)
> |                | Trigger word | No Trigger word | Total |
> |----------------|--------------|------------------|--------|
> | **Toxic**      | 0.19         | **0.31**             | 0.50   |
> | **Neutral**    | 0.01         | **0.49**             | 0.50   |
> | **Total**      | 0.20         | 0.80             | 1.00   |
>
> The Easy policy strongly prefers Neutral comments without Trigger words, almost eliminating the subgroup: Neutral with Trigger words. We hypothesize that the preference toward comments with no trigger words within the Toxic class is an artifact of strong class imbalance in the original dataset. These two factors essentially eliminate the spurious correlation between the comment class and the presence of a Trigger word, yielding high worst-group accuracy. Since one of the groups is severely underrepresented, the bias level remains high.

---

> > ### Comment · Reviewer_KrE1 · 2025-08-03
> >
> > Thanks for the rebuttal. Most of my concerns are addressed, and I will raise the score.

---

### Official Review · Reviewer_s6MD · 2025-07-01

**Rating:** 5
**Confidence:** 2

**Summary:**

This work is about selecting coresets which is a subset of high-quality data from the model's large scale training data, in that way, model's training on coresets can maintain the power but with less computational resources. However, when spurious association (which means models are classifying objects by things except the objects themselves) occurs, current random coresets selection (strong baseline) may not be robust, since empirical results (Figure 1) display that different subsets perform highly different. This work claims to be the first to perform analysis on how different selection may influence the dataset bias and downstream robustness.

**Dataset Code Accessibility:**

Yes

**Dataset Code Comments:**

The code is accessible on a well-known platform (Github). Detailed Readme File is included to help the readers to utilize it.

**Ethical Considerations:**

No, there are no or only very minor ethics concerns

**Final Justification:**

Good work

**Limitations Weaknesses:**

N/A, good work, from my point of view.

**Strengths Contributions:**

Good motivation: Comprehensive literature review and the motivation is necessary. In figure 1, clearly showing what the existing coreset selection methods are weak at.

Well-structured paper: This work is well-written, and easy for readers to follow and understand, even though for readers who do not familiar with the fields of coreset selection.

Future impact: Since this work is first time to provide the systematic analysis on this new field, the significance should be large enough.

---

> ### Author Rebuttal · Authors · 2025-07-31
>
> Thank you for your encouraging review!  We're glad that you found our paper to be based on a **good motivation**, and backed by a **comprehensive literature review**, especially to show what the existing coreset selection methods are weak at.
> It's also rewarding to hear that you found the paper to be **well-structured and easy for readers to follow and understand**, especially since unraveling nuanced, yet digestible insights from extensive experimental results was central to our work.
> Your perspective on the future impact of this work, especially as the **first systematic analysis on this new field**, is deeply appreciated.
>
> While you did not raise any specific concerns, we are actively working to improve the manuscript further by refining some of the figures and clarifying the writing, based on comments from other reviewers. We hope these revisions will make the work even more accessible and useful to a broader audience.

---

### Official Review · Reviewer_X77M · 2025-07-01

**Rating:** 5
**Confidence:** 1

**Summary:**

This paper conducts a review of existing coreset selection methods from the point of view of
group robustness. A large variety of datasets with known spurious features are used.
The experiments show how bias, robustness and accuracy change for different selection rates and policies.
Main conclusions are that embedding-based methods tend to be less biased than learning based ones,
low data bias does not necessarily lead to more robust classifiers, and that performance of small coresets
may differ significantly from larger ones.

**Additional Feedback:**

On Figure 3, is the "Group Balanced" line supposed to be the "R-Gbal"?

**Dataset Code Accessibility:**

Yes

**Dataset Code Comments:**

I didn't run the code, but it looks to be complete and sufficiently documented.

**Ethical Considerations:**

No, there are no or only very minor ethics concerns

**Final Justification:**

With the additional data and interpretations provided, the authors address all of the concerns that I have.

While I am not familiar with the field and can't fairly judge the importance of the contribution, the overall quality of the paper is high enough for me to recommend to accept it.

**Limitations Weaknesses:**

However, there are still some issues with the presentation that I would like to point out.

Most notably, tables and graph appear to be incomplete. For example, Table 1 doesn't report the scores about the
Forgetting method, but it is still discussed in the body of the paper. Also, selection Stratified and Median
selection policies are not considered in Section 4.2.
I feel like this data needs to be added for the work to be systematic.

Finally, I would like to mention that the formatting of the paper appear to be slightly
different from the NeurIPS template. In particular, I noticed that the lines are not numbered
and the spaces after section titles are slightly bigger.

**Strengths Contributions:**

The technical quality of this work is rather good. Experimental setups are described in detail and
the code is provided. The experiments for the most part are comprehensive. Being a review,
the paper might be helpful to other researchers in the area, as it facilitates the comparison of
new method against old ones, and challenges existing beliefs about the relationship between bias and robustness.

The introductory part is fairly well written.

---

> ### Author Rebuttal · Authors · 2025-07-31
>
> Thank you for your thoughtful and encouraging assessment.
> We appreciate the recognition of the **technical quality of our work** and the acknowledgment that **experiments are comprehensive and helpful to other researchers in the area**.
> We are particularly excited that you pointed out our analysis **challenges existing beliefs about the interplay between bias and robustness**, since it was one of the major goals of our work.
>
> ### 1. Forgetting, Uncertainty, and SupProto results corresponding to Table 1
>
> We appreciate your taking thorough notice of the tables and graphs.
> We presented only the most relevant results for each subsection in the main paper to preserve readability while presenting additional results in the appendix.
> Here we detail the additional results.
>
> In Table 1, we presented the worst-group accuracies for all datasets at a 40% selection rate for two methods representing the two main families of methods. EL2N for learning-based selections and SelfSup for embedding-based selections.
> The main observation of the results from Table 1, as mentioned in table caption, is that “In general, Difficult selection policies with EL2N scores yield robust classifiers. Across the results for SelfSup scores, no one policy stands out consistently.”
>
> Reported below are the worst-group accuracies for the rest of the methods: Forgetting and Uncertainty from the learning-based family, and SupProto from the embedding-based family. The selection rate is 40%, the highest performance for each dataset is bolded.
>
>
> Legend: (U) Uncertainty, (F) Forgetting, (S) SupProto
>
> | Dataset           | Diff (U) | Strat (U) | Med (U) | Eas (U) |   | Diff (F) | Strat (F) | Med (F) | Eas (F) |   | Diff (S) | Strat (S) | Med (S) | Eas (S) |
> |-------------------|----------|-----------|---------|---------|---|----------|-----------|---------|---------|---|-----------|-----------|---------|---------|
> | Cmnist            | 37.8     | **82.3**  | 1       | 0       |   | 9.9      | 4         | 0       | 0       |   | **87**    | 48.5      | 0       | 0       |
> | waterbirds        | 72.6     | 71.2      | 51      | 28.2    |   | **75.4** | **79.6**  | 53.8    | **52**  |   | 66.5      | 58.8      | 59.4    | 30      |
> | Urbancars_cooccur | **54.8** | **55.2**  | 46.4    | **50.4**|   | 51.2     | 54        | 31.6    | 31.6    |   | 46.8      | 50.4      | 50      | 44.8    |
> | Metashift         | **73.8** | 69.2      | **70.2**| 65.4    |   | 71.7     | **76.4**  | 46.6    | 48.7    |   | 67        | 73.8      | 72.3    | 58.5    |
> | Nico_95_spurious  | 44       | 42        | 36      | 30      |   | 44       | 40        | **46**  | 40      |   | 38        | 42        | 42      | 38      |
> | Urbancars_bg      | **52.8** | **53.2**  | 26      | 15.6    |   | 47.6     | 50.8      | **54**  | **54**  |   | 53.2      | 52.4      | 36.8    | 13.6    |
> | Urbancars_all    | 22.4     | **26.4**  | 8       | 7.2     |   | 20       | 23.2      | 16      | 16      |   | 20.8      | 19.2      | 17.6    | 5.6     |
> | MultiNLI          | **61.5** | **64.0**  | 53.8    | 49.2    |   | 59.5     | 57.8      | 52.4    | 52.4    |   | –         | 52.2      | –       | –       |
> | Civilcomments     | 60.2     | 64.2      | 61.0    | 41.5    |   | 60.9     | 65.3      | **82**  | 79.5    |   | **76.2**  | **77.1**  | 74.1    | 75.2    |
> | CelebAhair        | 42.2     | 42.2      | 62.8    | **78.9**|   | 27.2     | 42.8      | 64.4    | 64.4    |   | 50        | 59.4      | 63.9    | **82.2**|
>
> Similar to EL2N, Uncertainty and Forgetting (other scores in the learning-based family), when combined with policies that prioritize more difficult samples (Diff and Strat), frequently yields more robust classifiers.
> And similar to SelfSup results from table 1, no clear policy stands out consistently for SupProto.
> Since the patterns that we unravel in the main paper appear reliably frequent across the rest of the methods for sections 4.2 and 4.3, we used EL2N to represent the learning-based selection regime.
>
> Results for different selection rates correponsing to Table 1 are presented in Appendix C.2. We will add the additional results presented above in the appendix of camera-ready.
>
> ### 2. Stratified and Median policy results corresponding to Section 4.2
>
> In Section 4.2, we elaborate on how “Coreset bias level is not a consistent indicator of downstream robustness”.
> In more detail, we show that for small coreset sizes, the robustness of models for Difficult coresets is unintuitively low, despite the bias levels being the lowest out of all policies.
>
> Here we present the bias levels (and corresponding worst-group accuracies) for Random, Difficult, Stratified, Median, and Easy policies corresponding to 2%, 5%, 20%, and 40% selection rates for EL2N selection on two datasets: Waterbirds and Urbancars_cooccur. (Lower values for bias level are desired, and higher values for worst-group accuracy are desired)
>
> Results for Waterbirds
>
> | %   | R            | Diff         | Strat         | Med           | Eas           |
> |-----|--------------|--------------|---------------|---------------|---------------|
> | 2%  | 2.00 (36.6)  |**1.38** (15.9)  | 1.65 **(40.8)**   | 1.96 (31.2)   | 2.00 (20.1)   |
> | 5%  | 1.93 (43.3)  | **1.57** (31.9)  | 1.88 **(51.1)**   | 1.98 (42.7)   | 1.98 (24.8)   |
> | 10% | 1.89 (45.5)  | **1.64 (68.5)**  | 1.84 (67.4)   | 1.97 (47.2)   | 1.99 (29.1)   |
> | 20% | 1.91 (53.1)  | **1.75 (74.8)**  | 1.89 (74.4)   | 1.98 (42.7)   | 2.00 (42.2)   |
> | 40% | 1.90 (62.0)  | **1.87 (74.6)**  | 1.90 (73.1)   | 1.99 (50.5)   | 1.99 (30.2)   |
>
> Results for Urbancars_cooccur
>
> | %   | R            | Diff         | Strat         | Med           | Eas           |
> |-----|--------------|--------------|---------------|---------------|---------------|
> | 2%  | 1.92 (47.6)  | **1.35** (36.0)  | 1.59 (46.0)   | 1.97 **(47.6)**   | 2.00 (45.6)   |
> | 5%  | 1.91 (43.6)  | **1.49** (27.2)  | 1.76 (44.8)   | 1.99 **(47.6)**   | 2.00 (46.0)   |
> | 10% | 1.90 (48.4)  | **1.55 (53.6)**  | 1.88 (46.0)   | 1.98 (46.7)   | 2.00 (47.6)   |
> | 20% | 1.90 (48.8)  | **1.64 (56.4)**  | 1.87 (49.2)   | 1.98 (48.6)   | 2.00 (46.4)   |
> | 40% | 1.91 (51.0)  | **1.77 (54.0)**  | 1.89 (51.2)   | 1.99 (46.4)   | 1.99 (49.6)   |
>
> Difficult selection (Diff) consistently yields the least biased coresets (As established in Section 4.1, EL2N difficulty scores have a strong preference towards bias-conflicting samples.)
> However, when coresets are small (2%, 5%), the Diff policy fails to achieve high robustness across both datasets.
> Thereby, we show that low bias-level does not consistently correlate with higher downstream robustness.
> Our core claims of Section 4.2 are further strengthened by these results: in such settings, Stratified and Median policies, which are consistently more biased than Difficult, counter-intuitively yield higher robustness. We will include these results in the appendix.
>
> ### 3. Formatting of our paper
> We appreciate your attention to detail here. Our submission is single-blind; therefore, we used the “preprint” version to format (\usepackage[preprint]{neurips_2025}), which did not include line numbers. We have not applied any style changes, and we believe the spacing was automatically determined by the LaTeX formatting, given the lengths of our sections and sizes of our figures. We apologize for any inconvenience with missing line numbers and will make sure everything is consistent for camera-ready.
> (Thank you for pointing out the small inconsistency of naming the Group Balanced policy. We will fix this.)

---

> > ### Comment · Reviewer_X77M · 2025-08-01
> >
> > Thank you for providing additional results.

---

### Note · Authors · 2025-08-14

We thank all four reviewers for appreciating our work's strengths and contributions.
Based on their generous constructive feedback, we will update the final version of our paper as follows.

1. Adding the prescriptive suggestions as explained in the response to reviewer KrE1 to the main paper (we will use the additional page allowed for the camera-ready version for this).
2. Adding the overfitting and coverage analysis to the Appendix as further exploration into the small data regime
3. Extending Appendix C with additional results presented in the response to reviewer X77M

Thank you all again for your time!

---

### Decision · Program_Chairs · 2025-09-18

**Decision:**

Accept (poster)

**Comment:**

This paper presents the first comprehensive study on how coreset selection methods influence the group robustness of models. The authors analyze the impact of various sample scoring and selection policies across diverse datasets. The reviewers appreciated the paper's strong motivation, significant findings, and clear presentation. They also recognized the comprehensive experiments and the way the paper's findings challenged existing beliefs about the relationship between bias and robustness. While initial concerns were raised regarding incomplete tables and graphs, a lack of a theoretical foundation, and the analysis being limited to supervised learning for classification, the authors successfully addressed many of these issues through the rebuttal. In the end, the reviewers unanimously supported the paper. The AC concluded that the positive feedback and the successful rebuttal outweighed the remaining concerns, such as the absence of a theoretical foundation, and therefore agreed with the reviewers' consensus.